# Self-establishing communities enable cooperative metabolite exchange in a eukaryote

Kate Campbell[1,2], Jakob Vowinckel[1,2], Michael Mülleder[1,2], Silke Malmsheimer[1,2], Nicola Lawrence[3], Enrica Calvani[1,2], Leonor Miller-Fleming[1,2], Mohammad T Alam[1,2], Stefan Christen[4], Markus A Keller[1,2], Markus Ralser[1,2,5*]

[1]Department of Biochemistry, University of Cambridge, Cambridge, United Kingdom; [2]Cambridge Systems Biology Centre, University of Cambridge, Cambridge, United Kingdom; [3]The Wellcome Trust Gurdon Institute, University of Cambridge, Cambridge, United Kingdom; [4]Institute of Molecular Systems Biology, ETH Zürich, Zurich, Switzerland; [5]Mill Hill Laboratory, The Francis Crick Institute, London, United Kingdom

**Abstract** Metabolite exchange among co-growing cells is frequent by nature, however, is not necessarily occurring at growth-relevant quantities indicative of non-cell-autonomous metabolic function. Complementary auxotrophs of *Saccharomyces cerevisiae* amino acid and nucleotide metabolism regularly fail to compensate for each other's deficiencies upon co-culturing, a situation which implied the absence of growth-relevant metabolite exchange interactions. Contrastingly, we find that yeast colonies maintain a rich exometabolome and that cells prefer the uptake of extracellular metabolites over self-synthesis, indicators of ongoing metabolite exchange. We conceived a system that circumvents co-culturing and begins with a self-supporting cell that grows autonomously into a heterogeneous community, only able to survive by exchanging histidine, leucine, uracil, and methionine. Compensating for the progressive loss of prototrophy, self-establishing communities successfully obtained an auxotrophic composition in a nutrition-dependent manner, maintaining a wild-type like exometabolome, growth parameters, and cell viability. Yeast, as a eukaryotic model, thus possesses extensive capacity for growth-relevant metabolite exchange and readily cooperates in metabolism within progressively establishing communities.

*For correspondence: mr559@cam.ac.uk

## Introduction

All living cells possess a system for biochemical reactions, the metabolic network, which supplies cells with their necessary molecular constituents. The reactions participating in this network are highly conserved, so much so that all life is made up of a markedly similar set of metabolites (*Braakman and Smith, 2013*; *Caetano-Anolles et al., 2009*). The functionality of the metabolic system is bound to a series of transport reactions that facilitate the uptake of metabolites from the environment, as well as metabolite export. Metabolite export primarily occurs for the purpose of maintaining balance of the metabolic system ('overflow metabolism') and to maintain chemical and physical integrity of the metabolic network. This includes indiscriminate metabolite export through non-specific multi-drug transporters required in removal of toxic metabolites for cells (*Paczia et al., 2012*; *Piedrafita et al., 2015*). Co-growing cells can uptake the released metabolites and exploit their presence. Indeed, to a lower extent, metabolite export of metabolites can specifically occur for the purpose of establishing inter-cellular metabolic interactions in biosynthetic metabolism

**eLife digest** Life is sustained by an array of chemical reactions that is collectively referred to as metabolism. Some of these reactions break down complex substances to release energy and vital compounds, while others make new molecules from smaller building blocks.

Bacterial communities are regularly composed of heterogeneous species, several of which have lost one or more essential metabolic pathways. Nevertheless, these cells can still survive by making use of metabolic products released by their neighbouring cells.

Yeast are single-celled fungi that also form colonies and, as eukaryotes, they possess cells that are more similar to our own. However, in the laboratory, complementary metabolically deficient yeast cells do not survive when mixed together. It was presumed this is because yeast cells make only enough of each essential metabolite for themself, and so can't replace those that are missing from their neighbouring cells. Campbell et al. now challenge this view by finding that yeast cells release a variety of metabolites, they use these released metabolites in preference to making their own, and possess the capacity to grow on the basis of a non-cell-autonomous metabolism.

This discovery came with the design of a new experimental test to study metabolite exchange interactions. This method uses yeast cells that have one or more of their own metabolic genes disabled, and instead have a copy of these genes on small circular DNA 'mini-chromosomes' (called plasmids). The gene on the plasmid can compensate for the yeast having its own gene missing, and allows the cell to still make the metabolic product it needs to survive. However, as a single cell divides to form a colony, cells randomly lose these plasmids, leaving some of the cells deficient for a particular metabolite. These cells can only survive if the other cells in the colony export the missing metabolite in the quantity needed for growth. Using this test, Campbell et al. found that yeast cells can export missing metabolites at levels needed to sustain these emerging metabolic mutants. Additionally, these yeast communities could grow at levels comparable to other yeast without metabolic deficiencies. The resulting colonies also feature one of several different genetic and metabolic profiles, which change in response to the metabolite that is missing.

These findings demonstrate that yeast cells can exchange high amounts of metabolites, sufficient to form cooperative colonies, and as metabolite concentrations are not altered compared to normal cells, it is likely that exchange of metabolites is ongoing between neighbours in yeast communities. The additional discovery that yeast stop making metabolites when they can obtain them from neighbouring cells has implications for research. This is because many yeast genetic studies use metabolically deficient strains that are supplemented in culture with metabolites. Future work could address whether such supplementation has kept certain functions of metabolism hidden.

(*Nigam, 2015*; *Paczia et al., 2012*; *Yazaki, 2005*) and can lead to mutualistic situations in which cells profit from coexistence (*Foster and Bell, 2012*; *Oliveira et al., 2014*). In between species, mutually positive interactions can readily establish when exchange concerns an overflow metabolite, exemplified by yeast–algae interactions that can form on the basis of a $CO_2$ and sugar exchange (*Hom and Murray, 2014*), or between different cells of the same species or tissue, exemplified in tumours, when lactate produced in excess by one cell type is metabolised by another (*Bonuccelli et al., 2010*), or between neurons and glial cells that exchange sugar metabolites (*Bélanger et al., 2011*; *Volkenhoff et al., 2015*).

It is more difficult to assess whether metabolite exchange is indicative of non-cell-autonomous metabolism, when exchange concerns metabolites that are needed by both exchange partners, amino acids, and nucleobases for instance. Exchange of costly intermediates is associated with a significant risk, as exported metabolites can be lost through diffusion, chemical damage, or cheating (*Dobay et al., 2014*; *Oliveira et al., 2014*; *Wintermute and Silver, 2010*). Despite these constraints, exchange of intermediates is frequently observed within bacterial microbial communities. Many bacterial species lose essential biosynthetic pathways, disabling them from living autonomously, which may explain why more than 90% of bacteria cannot be cultivated in the absence of a community environment (*Costerton et al., 1994*; *Johnson et al., 2012*). The energetic benefit and selective advantage associated with non-autonomous cellular metabolism is often not clear but might involve, for example, the ability to reduce genome size which would in turn facilitate faster

proliferation. This may explain why bacteria frequently appear to cooperate in the biosynthesis of more costly and biosynthetically complex metabolites, such as aromatics (*Mee et al., 2014*).

While metagenomics has boosted the knowledge of metabolite exchange strategies in bacteria (*Blaser et al., 2013*; *Manor et al., 2014*; *Zelezniak et al., 2015*), relatively little is known about eukaryotic species. This includes yeast, a popular single cellular eukaryotic model organism, whose metabolic capacities are regularly exploited in biotechnology. Yeast cells are known to participate in multi-species communities (e.g. on human skin (*Findley et al., 2013*)), but as wild yeast isolates usually maintain similar prototrophic genomes (*Jeffares et al., 2015*; *Liti et al., 2009*), metagenomic data are not conclusive about yeast's metabolite exchange strategies. In laboratory experiments, yeast cultures were, however, not effective in supporting co-growth of auxotrophs that have complementary defects in amino acid and nucleotide metabolism, unless they were genetically modified to increase metabolite export (*Müller et al., 2014*; *Shou et al., 2007*). This contrasts with analogous studies in bacterial species, in which such growth experiments regularly show that co-cultured cells can overcome complementary metabolic deficiencies (*Foster and Bell, 2012*; *Oliveira et al., 2014*; *Pande et al., 2014*; *Vetsigian et al., 2011*). In the absence of quantitative metabolite data, this observation has triggered the conclusion that co-growing prototrophic yeast cells produce amino acid and nucleotide metabolites predominantly for themselves and export them at insufficient quantities to support growth of co-growing cells (*Momeni et al., 2013*; *Shou et al., 2007*).

Conflicting with this interpretation, we here report that yeast colonies maintain a rich exometabolome and that cells exploit this metabolic pool preferentially over their own biosynthetic capacities, which implies that metabolite exchange establishes as a natural property of yeast growth. To test whether yeast indeed possesses the capacity for metabolite exchange at growth relevant quantities, we established an alternative method to co-culture experiments. We exploited the stochastic segregation of episomes to randomly and progressively introduce metabolic auxotrophies into a yeast population which self-establishes from an initially prototrophic cell. This strategy enabled co-growing auxotrophs to enter an efficient state of metabolic cooperation, named self-establishing metabolically cooperating communities (SeMeCos). Despite an auxotrophic cell composition of up to 97%, SeMeCos achieve metabolic efficiency, growth parameters, and cell viability similar to that of genetically prototrophic cells, revealing a natural capacity of yeast to exchange metabolites at growth relevant quantities. In a SeMeCo that possesses auxotrophies in histidine, leucine, uracil and methionine metabolism, we distinguish up to eight cell types, each of which is unable to survive on its own, or in co-culture studies, however, could adapt effectively and cooperatively to overcoming metabolic deficiencies, once self-established in a community structure. Communities have a stable population composition as well as distinct spatial heterogeneity, which was, however, not essential for metabolite exchange, as SeMeCos maintain growth in liquid suspension. Self-establishing, complex communities thus demonstrate that yeast is not only exchanging metabolites, but is also able to do so at growth relevant quantities, to facilitate growth on the basis of a non-cell-autonomous metabolism.

## Results

### Yeast cells do not complement metabolic deficiencies in co-cultures but maintain the required exometabolome

Histidine, leucine, uracil, and methionine biosynthetic pathways were chosen for our study as (i) they can be interrupted by deletion of a single, non-redundant gene, which has been reported not to cause compensatory mutations and (ii) because cells possess efficient uptake mechanisms for these nutrients (*Mülleder et al., 2012*; *Pronk, 2002*; *Teng et al., 2013*). Paired combinations of histidine (*his3Δ*), leucine (*leu2Δ*), uracil (*ura3Δ*), or methionine (*met15Δ*) auxotrophs were unable to sustain growth in the absence of supplementation required for both individual cell types (*Figure 1A*). A similar result was obtained by co-culturing flocculating yeast cells (*Figure 1—figure supplement 1*), which are able to maintain biofilm-like physical contact (*Smukalla et al., 2008*), and in *Schizosaccharomyces pombe* (*Figure 1B*), indicating evolutionary conservation of this observation in yeast species.

In two previous studies, leucine/tryptophan and adenine/lysine auxotrophic cell pairs, respectively (*Müller et al., 2014*; *Shou et al., 2007*), could co-grow upon removing metabolic feedback control. Feedback resistance renders cells metabolite over-exporters, leading to the conclusion that wild-

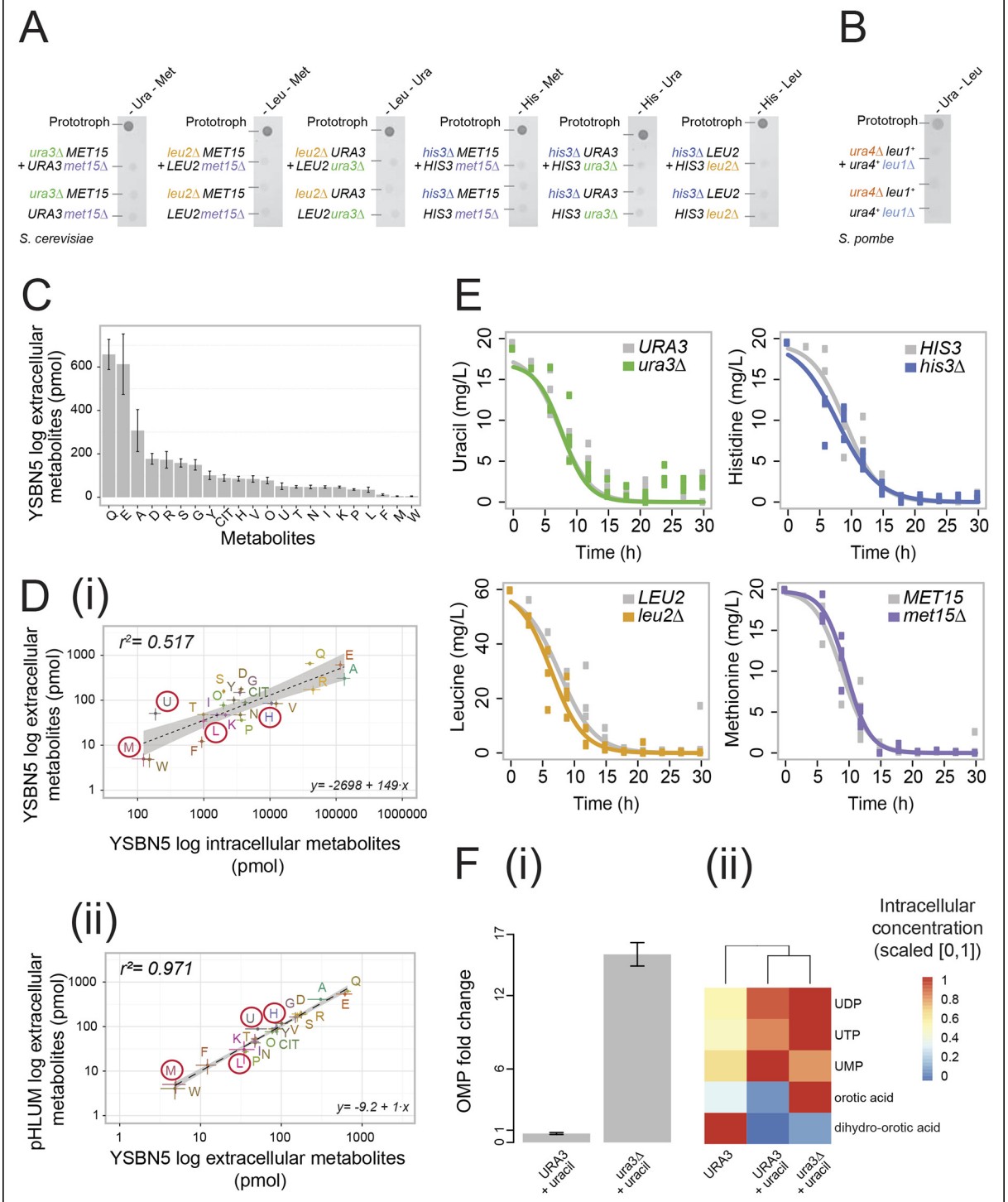

**Figure 1.** Yeast auxotrophs do not compensate for metabolic deficiencies upon co-culturing, yet export the relevant metabolites and prefer metabolite uptake over self-synthesis. (**A**) Complementary pairs of *Saccharomyces* cerevisiae auxotrophs do not overcome metabolic deficiencies upon co-culturing. *his3Δ, leu2Δ, met15Δ*,and *ura3Δ* yeasts were combined in complementary pairs and spotted on corresponding selective media. No pairs exhibited co-growth together. (**B**) A complementary pair of *Schizosaccharomyces pombe* auxotrophs does not overcome metabolic deficiencies upon co-culturing. *leu1Δ* and *ura4Δ* yeasts were combined in a complementary pair and spotted on corresponding selective media. No co-growth occurred. (**C**) The concentration of metabolites in the *S. cerevisiae* colony exometabolome obtained from 1.3e08 YSBN5 cells grown in a colony on synthetic minimal agar media (SM) and quantified by LC-MS/MS. Abbreviations: single letter IUPAC amino acid codes, O = ornithine, CIT = citrulline. n = 3, error bars = ± SD. (**D**) (**i**) Metabolites quantified as in (**C**), comparing intracellular (total cell extracts) and extracellular metabolite concentrations in YSBN5. n = 3, error bars = ± SD. Dashed line: linear regression fit, grey band shows 95% confidence region. (**ii**) Metabolites quantified as in (**C**), comparing

*Figure 1. continued on next page*

*Figure 1. Continued*

extracellular metabolite concentrations of YSBN5 and BY4741-pHLUM yeast colonies grown on minimal media. H, L, U, and M are highlighted in red circles. n = 3, error bars = ± SD. Dashed line: linear regression fit, grey band shows 95% confidence region. Abbreviations IUPAC codes; H = histidine, L = leucine, U = uracil, M = methionine. (E) Consumption of uracil, histidine, leucine, and methionine in yeast batch cultures in synthetic complete (SC) media as measured by LC-MS/MS. Uracil, histidine, leucine, and methionine prototrophic cells consume these metabolites at rates and quantities comparable to the corresponding auxotrophic strains. (F) (i) Deletion of *URA3* (Orotidine-5'-phosphate decarboxylase) causes accumulation of the Ura3p substrate orotidine-5'-phosphate (OMP), when cells are supplemented with 20 mg/L uracil (fold change of OMP abundance, relative to *URA3* without uracil supplementation), as determined by LC-MS/MS. Error bars = ± SD. (ii) Uracil supplementation of wild-type cells alters their metabolite profile to resemble *ura3Δ* cells, which obtain uracil solely from the growth media. Heatmap scaling ([0,1] and min, max per metabolite) was based on median concentration. The dendrogram was constructed by comparing euclidean distance (dissimilarity) between samples.

The following figure supplements are available for Figure 1:

**Figure supplement 1.** Flocculation does not enable *Saccharomyces cerevisiae* cells to establish viable co-cultures.

**Figure supplement 2.** Uracil biosynthetic genes in the uracil prototroph (*URA3*) and auxotroph (*ura3Δ*) remain expressed in (uracil supplemented) SC media, as determined by RNA sequencing.

type yeast cells produce intermediates primarily for themselves, at quantities that are not sufficient for growth relevant metabolite exchange (*Momeni et al., 2013*; *Shou et al., 2007*). In a detailed analysis of the intra-colony exometabolome, using an ultra-sensitive mass spectrometry method, the intra-colony fluid showed however to contain a plethora of metabolites, with the amino acids glutamine, glutamate, and alanine being the most highly concentrated (*Figure 1C*). Furthermore, histidine, leucine, methionine, and uracil all showed to be part of this exometabolome (*Figure 1C*). These measurements were obtained from cells in exponential growth phase, where apoptosis and necrosis are negligible. Comparing extracellular metabolite concentrations to intracellular levels (the endometabolome) we observed a general trend of correlation between the highest and lowest concentrated metabolites ($r^2$ = 0.517; *Figure 1Di*), but overall extracellular metabolite concentrations do not replicate the corresponding endometabolome. Tryptophan, phenylalanine, proline, and valine, for instance, were over-proportionally more concentrated inside the cell, whereas uracil, serine, tyrosine, and glycine were relatively over-represented in the extracellular fluid (*Figure 1Di*). Instead, highly similar exometabolome concentration values ($r^2$ = 0.971) were observed in the related yeast strain BY4741 upon complementing its auxotrophies with the centromere-containing single-copy vector (a minichromosome), 'pHLUM', which contains all four marker genes (*Mülleder et al., 2012*) (*Figure 1Dii*). Metabolite concentrations in the exometabolome between these two related yeast strains are hence substantially more similar than the endo- versus exometabolome in the same strain, implying that the intra-colony exometabolome is a distinct metabolite pool.

A second requirement to establish metabolite exchange is that cells need to be able to sense extracellular metabolites and to exploit them as a nutrient source. Yeast is known to uptake amino acids when they are available extracellularly (*Stahl and James, 2014*). We tested how extensive this uptake was by comparing the uptake rates between auxotrophs and prototrophs. Remarkably, prototrophic cells consumed histidine, leucine, methionine, and uracil at a comparable rate to the genetic auxotrophs, who depend 100% on external metabolite pools (*Figure 1E*). This demonstrated that yeast cells completely shift from *de novo* synthesis to uptake in the presence of each of the four metabolites. Studying the *URA3* genotype in greater detail confirmed the preference of uptake over self-synthesis. Enzymes involved in uracil biosynthesis remained expressed in both the *URA3* and the *ura3Δ* strains under fully supplemented conditions (*Figure 1—figure supplement 2*), but uracil biosynthesis-related intermediates shifted to similar concentrations both in the wild-type strain and in the *ura3Δ* strain once uracil was supplemented (*Figure 1F*). The only exception was the direct substrate of the *URA3* enzyme (orotidine-5'-phosphate decarboxylase), orotidine-5'-phosphate (OMP), which accumulated upon uracil supplementation once its metabolising enzyme (*URA3*) was deleted (*Figure 1Fi*). In summary, yeast cells do not compensate for metabolic deficiencies in co-culture experiments consistently as others reported previously (*Müller et al., 2014*; *Shou et al., 2007*), but they (i) export the relevant metabolites even when grown on minimal media and (ii) take up histidine, leucine, uracil, and methionine at similar rates to auxotrophs if supplementation is available. At least

for uracil, (iii) the biosynthetic enzymes and majority of biosynthetic intermediates in the supplemented wild-type cell resemble those of the corresponding auxotroph.

## Yeast can enter a state of efficient metabolic cooperation within a self-establishing community

In light of these results, we speculated that the inability to cooperate could be found in the nature of the co-culturing experiment. To establish an alternative method, we made use of a, in other circumstances disadvantageous, property of yeast plasmids, their occasional, stochastic loss from cells (segregation). Segregation is observed for both popular replication types, centromeric 'cen' and 2µ, at a rate of 2–4% expressed per cell division (*Christianson et al., 1992*). This property allowed us to randomly and progressively introduce auxotrophies into a developing yeast community starting from a single, initially prototrophic, cell: when a plasmid carries a gene that complements for an auxotrophy, a newly budded cell re-gains the metabolic deficiency according to the segregation of its plasmid. We transformed plasmids from the classic pRS and p400 series which express *HIS3*, *LEU2*, *MET15*, or *URA3* genes under the respective *S. cerevisiae* promoters (*Christianson et al., 1992*; *Mumberg et al., 1995*; *Sikorski and Hieter, 1989*) into the standard laboratory strain BY4741, deficient in these markers (*Brachmann et al., 1998*) (*Figure 2Ai*). As expected, the transformed cells grew competently in the absence of histidine, leucine, uracil, and methionine supplementations. We then quantified plasmid segregation and confirmed earlier literature values (*Figure 2Aii* and *Figure 2—source data 1*) (*Christianson et al., 1992*; *Ghosh et al., 2007*).

Cells having lost prototrophy can only continue growth if they obtain the relevant nutrient from the environment (*Figure 2B*). Transferred to minimal media, the lack of nutrient supplementation leads to three possible outcomes (*Figure 2C*): First, if the cooperative potential would not suffice to overcome the increasing content of metabolically deficient cells, colony growth would only be explained by cells maintaining all four plasmids (*Figure 2C left*). Alternatively, if the segregation is faster than the growth rate of cells carrying four plasmids, the colonies would not be able to grow (*Figure 2C centre*). Finally, the third outcome is that colony growth continues, despite an increasing auxotrophic composition, facilitated by cells exchanging histidine, leucine, uracil, and methionine at growth relevant quantities (*Figure 2C right*). First, we observed that upon approximately 33 biomass doublings, segregation for *HIS3* and *URA3* had continued until less than 50% of cells were prototrophs (*Figure 2D*), even though a 1:1 co-culture of the same auxotrophs was not able to co-grow (*Figure 1A*). Then, we assayed for the formation of a heterogeneous yeast community, starting from the four-plasmid (4P) strain, that can give rise to the emergence of 16 complementary auxotrophic genotypes (*Figure 2B*). In the 4P strain, individual plasmid segregation rates were similar but not identical to yeast carrying one plasmid at a time and were in linear correlation, indicating that no specific interaction between the plasmids occurred (*Figure 2Aii*). With a total segregation rate of 11%, 4P cells regain auxotrophy rapidly so that only 21 cell divisions (doublings) would result in >90% of cells losing prototrophy (*Figure 2E*). The continuous loss of prototrophy from the 4P strain was experimentally confirmed on rich (YPD) media; In the presence of rich supplementation, only 45 biomass doublings resulted in 96% of cells losing prototrophy (*Figure 2F*).

Testing whether cells can maintain growth by cooperating in the biosynthesis of histidine, leucine, uracil, and methionine, colonies were grown over 7 days on minimal media agar through dilution and re-spotting once giant colonies had formed (every 48 hr), so that the continuous gain in biomass necessitates constant *de novo* synthesis of intermediate metabolites. The experiment yielded viable colonies and from the obtained biomass, we calculated that 57 doublings had occurred. Fifty-seven doublings would have been sufficient for >99% of cells to segregate (*Figure 2E*). Replica plating revealed a predominantly auxotrophic composition of the obtained colonies. These were composed of 73.3 ± 3.7% auxotrophic cells (*Figure 2G left*), of which 39.9% had lost one plasmid, 20.6% two, 6.6% three, and 6.1% had lost all four markers (*Figure 2G centre*). No auxotrophies were in a 1:1 ratio with each other (36.9% for uracil, 27.7% leucine, 23.7% histidine, and 11.7% methionine (*Figure 2G right*), despite the segregation rates predicting a relatively equal distribution, implying that selection pressure for certain metabotypes affected colony composition (*Figure 2G right* and *Figure 2—figure supplement 1*).

We also confirmed that *S. pombe* is capable of forming similar communities, indicating conservation in these evolutionary distant yeast species (*Figure 2—figure supplement 2*). As additional controls, we (i) re-isolated the auxotrophs from the established colonies, and repeated the co-culture

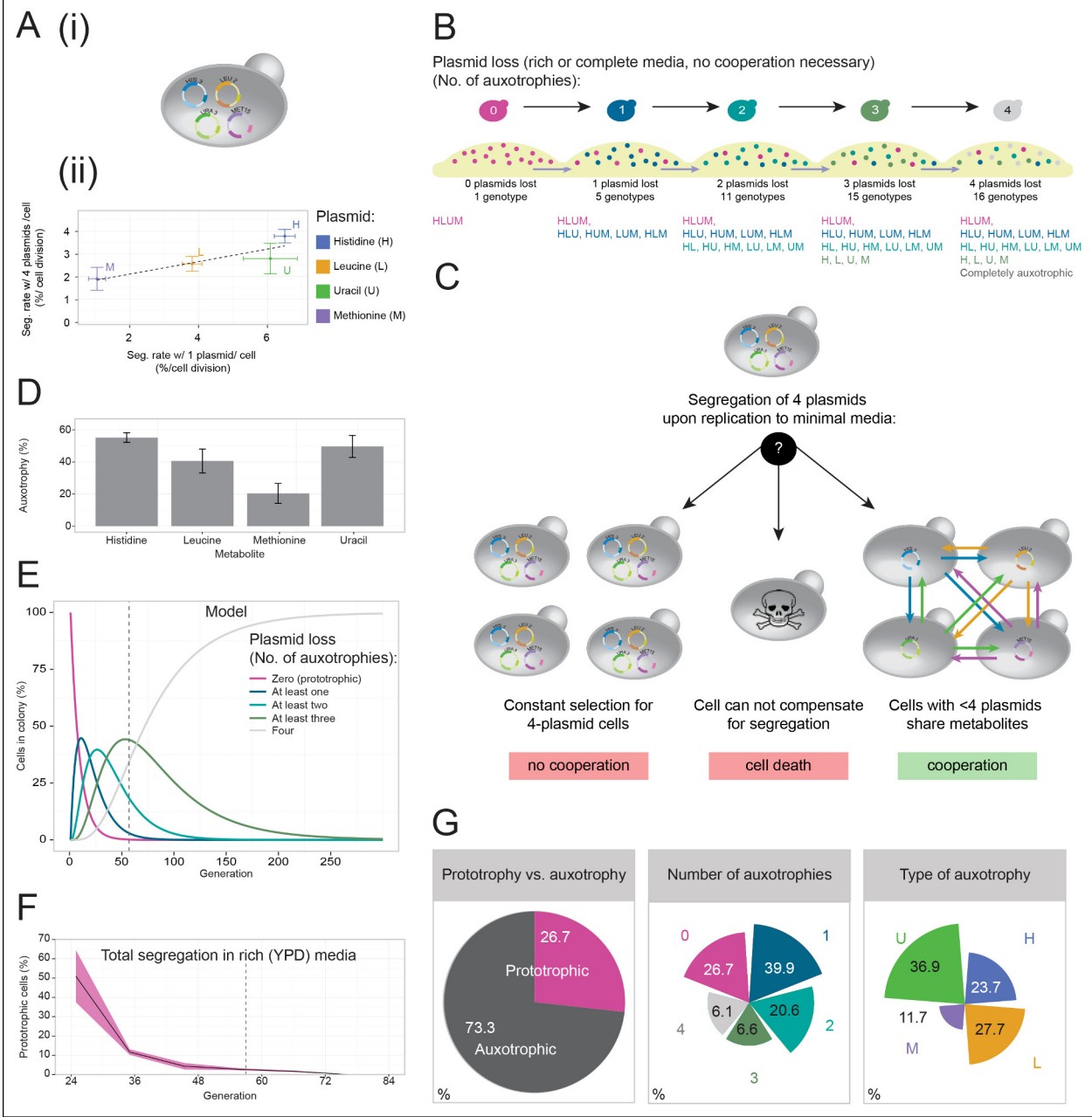

**Figure 2.** A self-establishing yeast community can cooperatively compensate for progressive loss of prototrophy on minimal media. (**A**) (i) Schematic illustration of BY4741 carrying four plasmids to complement its auxotrophies in histidine (*his3Δ1*), leucine (*leu2Δ0*), methionine (*met15Δ0*), and uracil (*ura3Δ0*). (ii) Plasmid segregation rates (probability of plasmid loss per cell division) of BY4741 carrying four plasmids encoding *HIS3* (p423), *LEU2* (pRS425), *URA3* (p426), and *MET15* (pRS411) (*y-axis*) compared to BY4741 carrying one plasmid at a time (*x-axis*). n = 3, error bars = ± SD. Dashed line: linear regression fit. (**B**) Schematic illustration of the segregate strain composition over time on rich or complete media where no cooperation is necessary for cells to survive. Sequential plasmid loss leads to an increase in auxotrophy, with loss of up to four plasmids leading to the formation of 16 cell types with varying metabolic capacity (metabotypes). (**C**) Three possible outcomes for BY4741 carrying four segregating plasmids, when establishing a colony on minimal media; (i) no cooperation, only cells carrying four plasmids grow, (ii) no cooperation but plasmid segregation is faster than the growth rate of cells carrying four plasmids leading to no growth capacity. Finally (iii), cells cooperate, wherein cells that have obtained auxotrophy continue growth by sharing metabolites with neighbouring cells in the colony. (**D**) Auxotrophy of BY4741 colonies carrying single plasmids encoding *HIS3* (p423), *LEU2* (pRS425), *URA3* (p426), and *MET15* (pRS411) on selective media after approximately 33 doublings. The number of plasmid-free cells (% auxotrophy abundance) was measured by replica plating. n = 3, error bars = ± SD. (**E**) Mathematical simulation of segregation over time, starting from 100% cells carrying four plasmids, based on the experimentally measured segregation rates. Highlighted is the situation after 57 doublings (achieved in dashed line) where >99.9% of cells have segregated >1 plasmid. (**F**) Segregation over time in a colony on rich media (no selection to maintain the plasmids); starting from a micro-colony of four-plasmid prototrophic cells on minimal media, cells were transferred to rich

*Figure 2. continued on next page*

*Figure 2. Continued*

(YPD) media and established as a giant colony, segregation was followed by replica plating. Biomass gain is counted from the single cell. (**G**) Giant colonies established for 57 biomass doublings on minimal media are composed of (*left*) 73.3% auxotrophic cells, (*centre*) contain a mixed number of auxotrophies and (*right*) a non 1:1 ratio of auxotrophy types. (n = 542 genotyped cells). Colony growth is achieved, despite the majority of cells possessing one or more auxotrophies.

The following source data and figure supplements are available for figure 2:

**Source data 1.** Plasmid segregation rates.

**Figure supplement 1.** Experimentally obtained colony composition, compared to the composition expected if segregation continued without selective pressure to maintain cells able to synthesise leucine, uracil, methionine and histidine.

**Figure supplement 2.** *Schizosaccharomyces pombe,* like *Saccharomyces cerevisiae,* are also able to establish SeMeCo colonies.

**Figure supplement 3.** Complementary pairs of auxotrophs, re-isolated from established SeMeCo colonies, do not overcome metabolic deficiencies upon co-culturing, similar to the original strains .

**Figure supplement 4** . Different auxotrophy combinations do not enable metabolic cooperation.

experiment, after having grown the cells for 48 hr in supplemented media, as with the original strains (*Figure 1A*). Even when isolated from a functional cooperating colony, complementary auxotrophic cells did not complement each other's deficiencies upon co-culturing (*Figure 2—figure supplement 3*), ruling out the possibility that new mutations altering metabolite exchange capacities could explain the formation of the cooperating community. We also mixed all four auxotrophs together, both in a 1:1 mixture, as well as in the ratio observed from the community and performed co-culturing both with and without co-cultivation before spotting; These attempts did not result in successful co-growth either (*Figure 2–figure supplement 4*). Hence, by exploiting plasmid segregation to overcome culturing and allowing the community to self-establish, heterogeneous yeast colonies were formed, which could sustain exponential growth under nutrient limitation, despite the majority of cells being auxotrophic for at least one metabolite. These co-growing cells could therefore overcome metabolic deficiencies through cooperative metabolism, demonstrating that yeast possesses metabolite exchange capacities at a growth relevant quantity.

## Self-established Metabolically Cooperating yeast populations ('SeMeCo') achieve wild-type-like metabolic efficiency

The obtained colonies were viable on minimal media and showed no apparent growth defects, despite containing a content of 73% auxotrophs, each of which were non-viable in co-culture studies (*Figure 1A,B*, *Figure 1—figure supplement 1*, *Figure 2—figure supplement 3,4*). To characterise the properties of this community, we started with LC-MS/MS to compare its exometabolome against prototrophic yeast strains (YSBN5, BY4741-pHLUM), and the unpassaged strain carrying the four plasmids (4P); (*Figure 3A,B* and *Figure 3—source data 1*). SeMeCo colonies possessed similar extracellular metabolite concentrations to prototrophic controls (*Figure 3Bi*). Of particular note are the extracellular concentrations of H, L, U, and M. Aside from a statistically non-significant trend towards a lower leucine concentration, only uracil (U) was significantly affected. To our surprise, however, the concentration of this metabolite was increased, indicating that SeMeCo had adapted by maintaining a higher level of uracil in its exometabolome (*Figure 3Bii*).

SeMeCo also maintained wild-type like growth efficiency and biomass-forming capacities under nutrient limitation (*Figure 3C*). Comparing SeMeCo against prototrophic yeast strains (*Figure 3A*), dry biomass formation did not vary significantly (*Figure 3C centre*; p-values = 0.27, 0.70, and 0.09 for FourP, pHLUM, and YSBN5, respectively). In liquid media, lag phase was prolonged, and the maximum specific growth rate (µmax) was slightly reduced compared to the genetically prototrophic YSBN5 or pHLUM cells (0.17 $OD_{595}$/hr vs 0.20 and 0.21 $OD_{595}$/hr, respectively) (*Figure 3C right*). However, this difference appeared more as a cost of plasmid segregation, as both lag phase and µmax did not vary significantly between SeMeCo and the FourP strain (0.17 and 0.16 $OD_{595}$/hr,

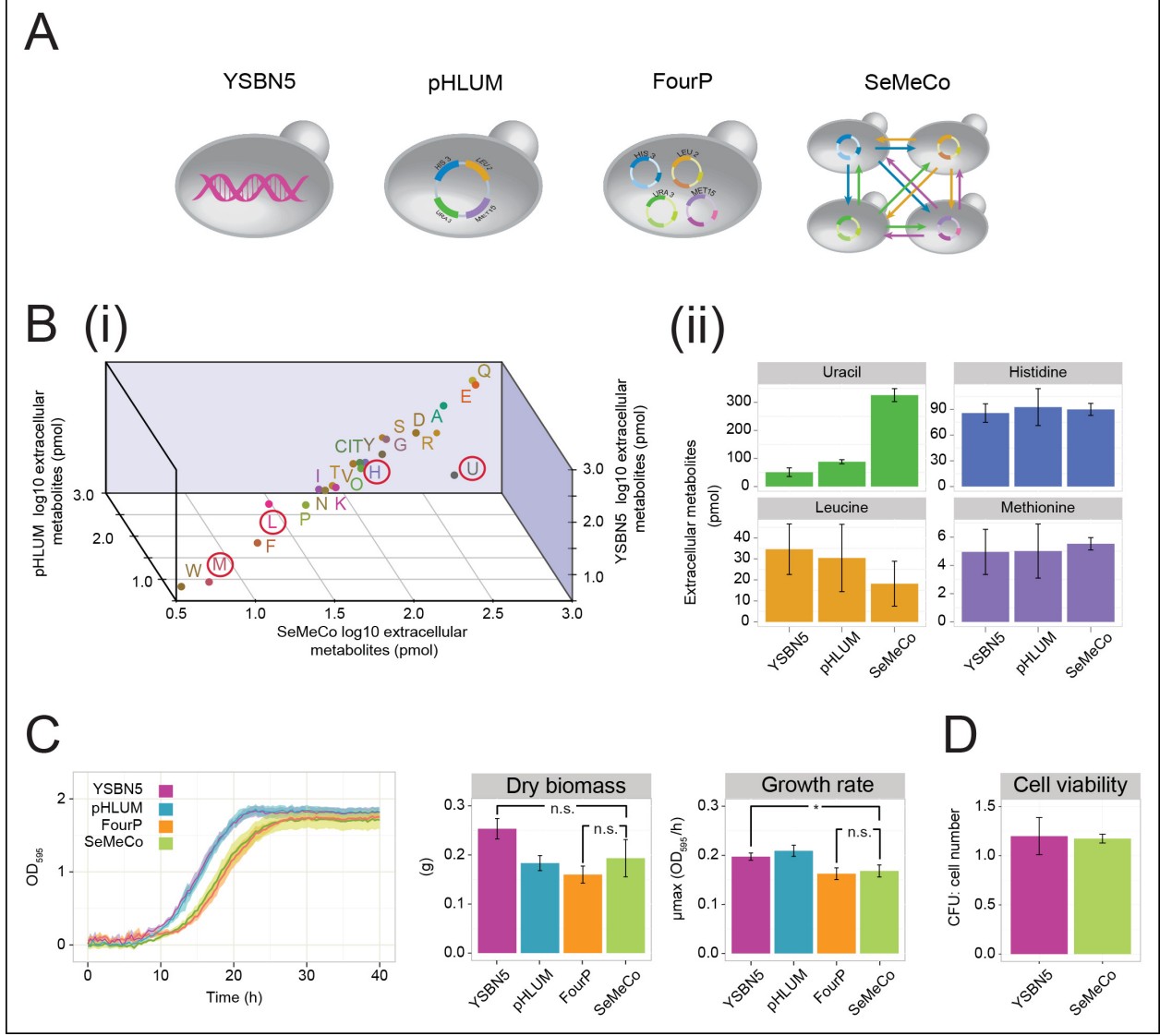

**Figure 3.** Growth and physiological parameters of the self-established metabolically cooperating yeast community 'SeMeCo'. (**A**) Schematic illustration of colonies derived from the genomically prototrophic yeast strain YSBN5, the single-vector complemented BY4741-pHLUM ('pHLUM'), BY4741 complemented with four plasmids ('FourP'), and the self established yeast population (SeMeCo; self-established metabolically cooperating yeast community); (*from left to right*). (**B**) (i) Extracellular concentrations of metabolites in colonies of YSBN5, pHLUM and SeMeCo growing exponentially on minimal media as determined by LC-MS/MS, n = 3. Histidine (H), leucine (L), methionine (M), and uracil (U) are highlighted in red circles. (ii) Detailed extracellular concentration values of uracil, leucine, methionine, and histidine as determined by LC-MS/MS. n = 3, error bars = ± SD. (**C**) (*left*) Growth curve of YSBN5, pHLUM, FourP, and SeMeCo as determined by measuring optical density ($OD_{595}$). n = 3, error area = ± SD. (*centre*) Dry biomass collected from 100 mL batch cultures after three days growth in minimal media, 30°C, n = 3, error bars = ± SD. (*right*) Maximum specific growth rate (μmax) as determined from $OD_{595}$ growth curves using a model-richards fit (*Kahm et al., 2010*). n = 3, error bars = ± SD. (**D**) The ratio of colony-forming units (CFUs) to number of cells used for plating, for YSBN5 and SeMeCo. n = 3, error bars = ± SD.

The following source data is available for Figure 3:

**Source data 1.** Absolute quantification of amino acids and uracil in yeast strains YSBN5, pHLUM and SeMeCo, absolute concentration values.

respectively) (*Figure 3C right*). Finally, we tested to what extent cell death occurs in the cooperating community. Both a wild-type (YSBN5) and a SeMeCo culture were grown to exponential phase and cells were counted. Then, the cultures were plated on SC media and the number of colony-forming units (CFUs) determined. The CFU count was nearly equal between SeMeCo and YSBN5, and similar to a 1:1 relationship to the cell count measured prior to spotting (*Figure 3D*). This indicates that cells

in SeMeCo have a comparable colony-forming capacity to that of exponentially growing wild-type cells, and in both populations, virtually every cell can form a new colony.

## SeMeCos reveal composition dynamics in response to nutritional changes

To establish if cells cooperating in SeMeCo are distributed in a random or organised manner, we analysed colony spatial structure using confocal fluorescence microscopy (*Figure 4A*). For this, the community was re-established with alternative plasmids that express the fluorescent protein markers CFP (cyan fluorescent protein), Venus (yellow fluorescent protein), Sapphire (a UV-excitable green fluorescent protein [*Sheff and Thorn, 2004*]), and mCherry (red fluorescent protein) coupled to the auxotrophic markers *HIS3*, *URA3*, *LEU2*, and *MET15*, respectively (*Bilsland et al., 2013*). Segregation of the labelled plasmids were within the same range, although not identical to the original pRS and p400 plasmids (*Figure 4—figure supplement 1*). Images were acquired with a SP5 on a DMI6000 inverted microscope (Leica, Wetzlar, Germany) and show the underside of a live two day micro-colony which had, prior to imaging, been growing on minimal media (SM). In our hands, the Sapphire-*LEU2* fluorescence was also visible under the imaging conditions used to visualise Venus-*URA3*. For this reason, the Venus-*URA3* channel was removed from the colony image. The spatial heterogeneity of fluorescent markers in the micro-colony revealed that cells in SeMeCo unequally distribute over the macroscopic structure, and form regions where the biosynthesis of a particular metabolite dominates (*Figure 4A*). A prediction in truly cooperating communities is, however, whether complementary cells maintain physical proximity to each other, to oppose diffusion of exchanged metabolites (*Müller et al., 2014*). Using computational image analyses of colony micrographs, we find that even when the most stringent cut-off was applied, complementary metabotypes across the community maintained an average distance (6.86 μm) of less than two cell diameters (*Figure 4B*). Cells are hence most likely to exchange the majority of metabolites with those maintaining close proximity. Despite these results, SeMeCo could, however, continue growth after disruption of this spatial structure in liquid media (*Figure 3C*). To verify this assumption, SeMeCo was replicated for 7 days in liquid minimal media, as previously, with re-dilution every 2 days. Indeed, the liquid culture maintained a similar content of auxotrophs as obtained with colony grown SeMeCos (*Figure 4—figure supplement 2*). The capacity of SeMeCos to overcome metabolic deficiencies through metabolic cooperation is hence not in essence bound to colonial growth.

To determine not only the spatial but also the population structure, we switched back to the non-fluorescent SeMeCo to avoid confounding effects of fluorescent protein expression, and quantified by replica plating the colony contribution of all 16 possible metabotypes, resulting from all possible combinations of the four auxotrophies (*Figure 2B*). These experiments revealed that within SeMeCo, 95.6% of cells belonged only to 8 of the 16 possible metabolic combinations (*Figure 5A*). We questioned whether this composition was the result of a stochastic event, however, the dominance of the same metabotypes establish three times independently. Moreover, the eight successful metabotypes contributed to SeMeCo at comparable percentages (*Figure 5A* inset). Using our segregation rate model as well as growth rate data, we could rule out this colony composition being a result of (i) varying plasmid segregation rate, (ii) the number or type of auxotrophy, or (iii) differences in growth rates. First, a community composition calculated on the basis of the experimentally determined plasmid segregation values (*Figure 2E*, *Figure 2—figure supplement 1*) showed zero correlation with the actual population composition ($r^2 = 0.051$) (*Figure 5B*). Second, all histidine, leucine, uracil, or methionine auxotrophies, as well as all plasmid numbers (1 to 4) were found amongst both the frequent and rare metabotypes. For instance, while single uracil (*HIS3, LEU2, MET15, ura3Δ*; 19.7%) or leucine (*HIS3, URA3, MET15, leu2Δ*; 11.1%) auxotrophs were amongst the most frequent cells, their methionine-deficient counterparts (*HIS3, URA3, LEU2, met15Δ*; 0.4%) were among the most rare (*Figure 5A*). Also, the high frequency of the dual auxotrophs *LEU2, MET15, his3Δ, ura3Δ* (8.1%) and *HIS3, MET15, leu2Δ, ura3Δ* (10.1%), contrasts with the rareness of the other dual auxotrophs (*HIS3, URA3, leu2Δ, met15Δ* (1.1%), *MET15, URA3, leu2Δ, his3Δ* (0.2%), *URA3, LEU2, his3Δ, met15Δ* (0.5%), and *LEU2, HIS3, ura3Δ, met15Δ* (0.6%)). Thus, the number of plasmids or type of auxotrophy a cell had did not indicate whether a cell-type would be rare or frequent (*Figure 5A*). Finally, the growth rate of 16 strains, carrying the same marker and supplement combination that replicates the 16 metabotypes (*Mülleder et al., 2012*) did not distinguish the depleted from the selected metabotypes either (*Figure 5C*).

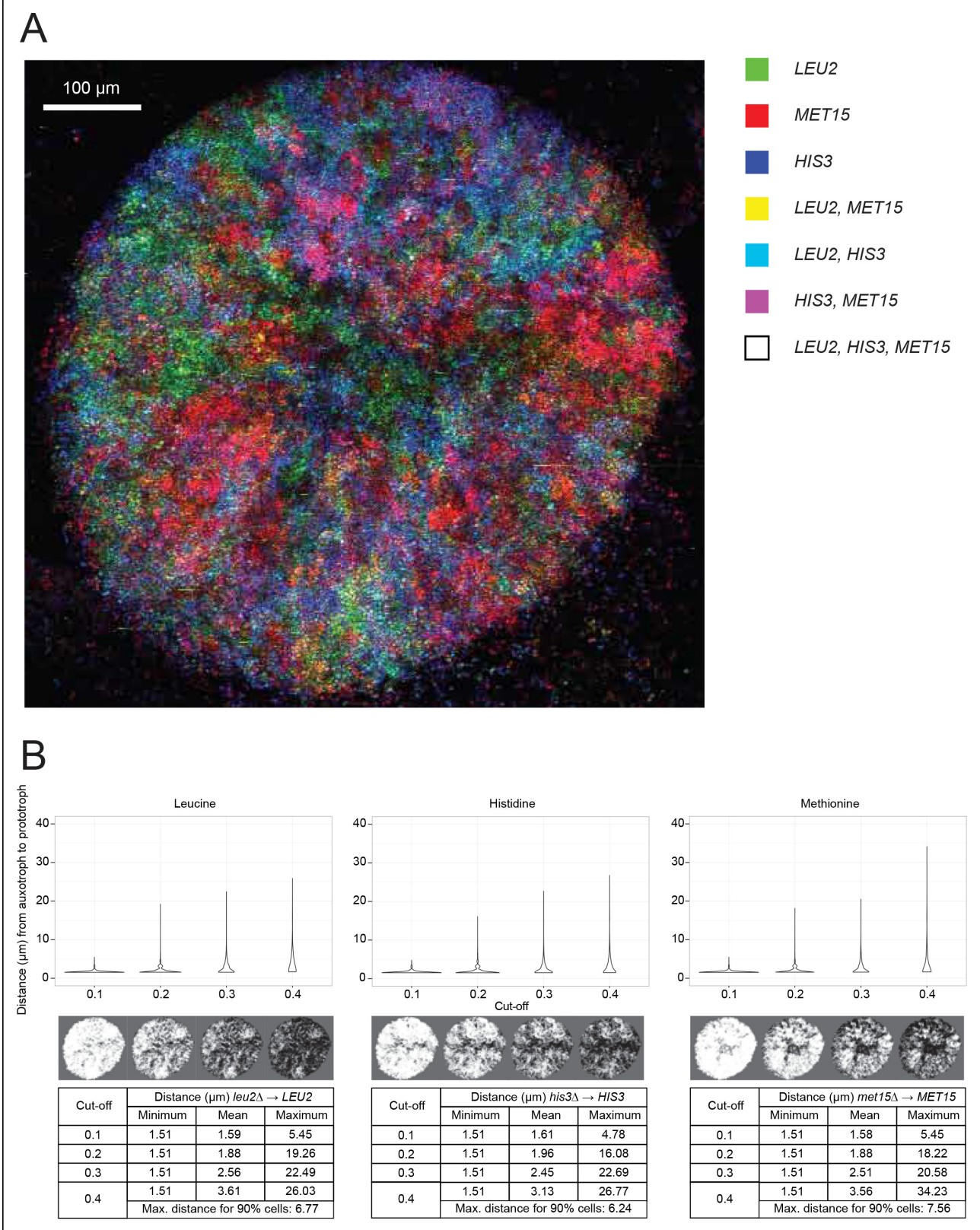

**Figure 4.** Spatial organisation of SeMeCo. (**A**) Spatial organisation of metabolically cooperating yeast micro-colony on minimal agar media (SM). SeMeCo was established with plasmids expressing fluorescent protein coupled to each auxotrophic marker; *LEU2*, *MET15*, and *HIS3* cells are coloured green, red, and blue, respectively. Cells containing more than one marker are coloured as a product of the additive RGB colour model. Two–day-old

*Figure 4. Continued*

live and growing micro-colony is visualised from underneath. (**B**) Minimum, mean and maximum distances between leucine, histidine, and methionine auxotrophs and their corresponding prototrophs in a SeMeCo colony. Maximum distance between auxotroph and prototroph for 90% of cells shows an average distance of 6.86 µm, using the highest cut-off. Despite the heterogeneous macroscopic colony composition, complementary auxotrophs are maintained in physical proximity to each other.

The following source data and figure supplements are available for figure 4:

**Source data 1.** Segregation rates of fluorescent protein plasmids from the yEp, pRS and p400 series.
**Figure supplement 1.** Plasmid segregation rates of fluorescent protein plasmids (%; probability of plasmid loss per cell division) of BY4741 carrying plasmids encoding *HIS3* (yEpCFP_HIS), *LEU2* (yEpSapphire_LEU), *URA3* (yEpVenus_URA), and *MET15* (pRS411-GPD*pr*-mCherry) respectively, compared to BY4741 carrying all four at the same time. n = 3, error bars = ± SD.
**Figure supplement 2.** SeMeCos continue growth in minimal (SM) liquid culture.

As growth potential and segregation parameters did not explain the population architecture of SeMeCo, we conclude that this community was selected for on its ability to cooperate effectively. If this interpretation is correct, it would imply that a different pressure to cooperate would result in a different SeMeCo composition. To test this hypothesis, we focussed on uracil, as mass spectrometry had detected an increase in uracil concentration in the SeMeCo colony exometabolome, indicating that uracil is the most limiting metabolite (*Figures 2G right* and *3Bii*). SeMeCos established on uracil adapted a different composition, resulting from a dramatic decline in cells needed to produce uracil. Importantly, this included the prototroph with its total content in the community decreasing from 26.7% in the original SeMeCo to solely 3.0%, so that 97.0% of cells were cooperating auxotrophs (*Figure 5D*). Hence, SeMeCo colonies establish a population that is dynamic to changes in the external metabolite pool, and can persist in a state with virtually all cells being genetically auxotrophic for at least one essential metabolite.

## Discussion

Metabolic exchange interactions occur frequently among cells that grow in proximity to one another, as metabolites are constantly released from cells for different reasons, such as overflow metabolism, metabolite repair, as well as export to facilitate metabolite exchange. In bacteria, a subset of such metabolite exchanges are of a cooperative nature in the sense that all exchange partners profit from this situation (*Oliveira et al., 2014*), whereas for the majority of eukaryotic organisms, metabolite exchange strategies remain unclear. Despite yeast auxotrophs being viable in supplemented and rich media (*Mülleder et al., 2012*), in the absence of amino acid supplementation, they fail to complement metabolic deficiencies in several pairs or higher order co-culture experiments (*Figure 1A,B*, [*Müller et al., 2014*; *Shou et al., 2007*]), a clear difference to bacterial studies, where similar experiments were effective (*Foster and Bell, 2012*; *Freilich et al., 2011*; *Harcombe, 2010*; *Pande et al., 2014*; *Ramsey et al., 2011*; *Vetsigian et al., 2011*). This led to speculations that yeast might, in contrast to many bacterial species, lack the required export capacities to enable growth relevant exchange of intermediary metabolites such as amino acids and nucleobases (*Shou et al., 2007*). Analysing the intra-colony exometabolome we could, however, detect the required metabolites; in fact, we found that cells within a colony are surrounded by a rich exometabolome. We also found that yeast would efficiently exploit the nutrients when available, to the extent that they solely rely on these extracellular metabolites. This result implied that metabolite exchange among co-growing yeast cells is frequent by nature; the lack of complementation in the co-culture experiments could thus reflect a limit of the experiment itself, and not represent the metabolite exchange capacities of yeast cells.

To circumvent combining two or more cultures, we chose an approach of synthetic biology and exploited the stochastic loss of plasmids to progressively introduce the metabolic deficiencies in random combination from an initially single cell. The progressive loss of prototrophy allowed cells to maintain cell growth on the basis of metabolite exchange, resulting in a community with 73% auxotrophy, which increased to 97% upon supplementation with the most limiting metabolite, uracil.

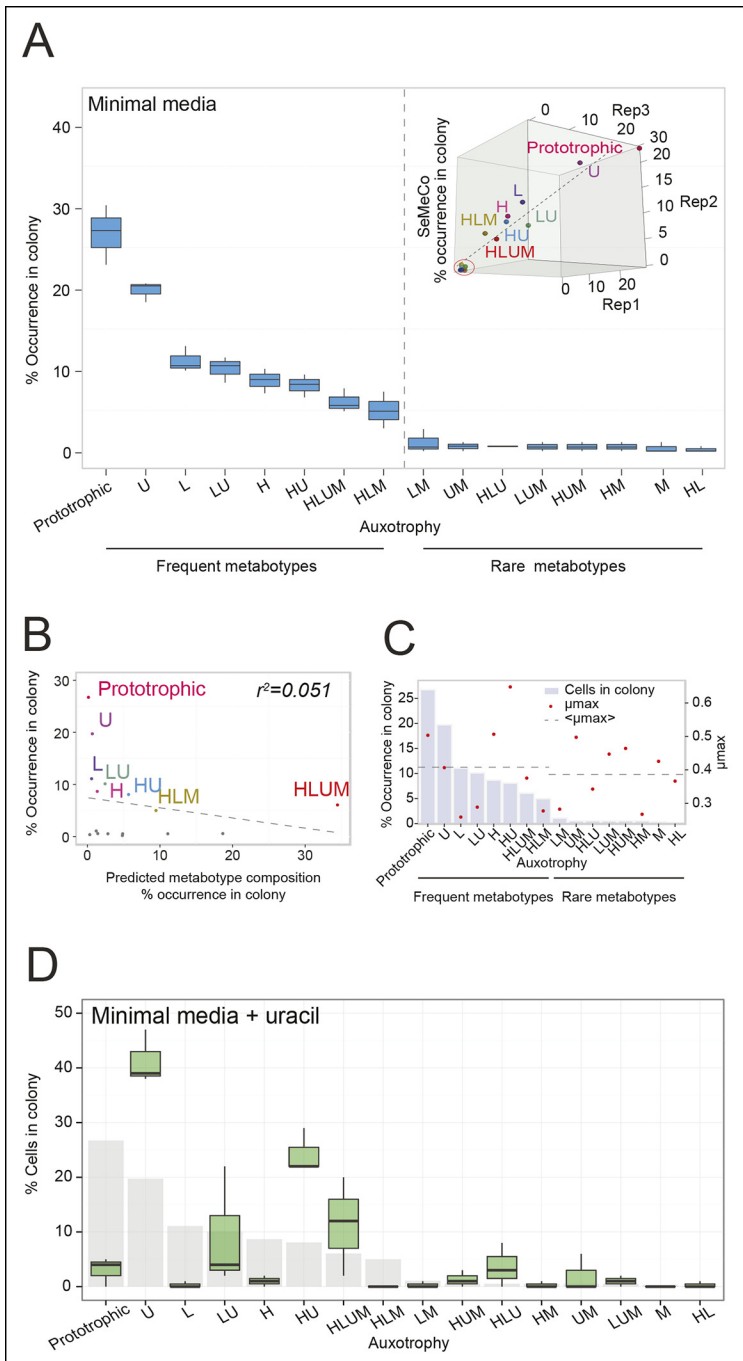

**Figure 5.** The community composition is distinct and dynamic. (**A**) Frequency of the 16 metabotypes that result from combination of histidine, leucine, methionine, and uracil auxotrophies as found within SeMeCo colonies. Separately established populations (n=3) were genotyped (>180 cells per colony) and eight metabotypes showed to dominate in the populations (*inset*). Frequency of the 16 metabotypes in independently established cell populations. The eight metabotypes of low frequency, which were depleted in all experiments, are highlighted with a red circle. (**B**) No correlation shown between the frequency of the 16 metabotypes in SeMeCo versus a segregation rate-predicted colony composition. Coloured points correspond to the experimentally observed eight frequent metabotypes. Dashed line: linear regression fit. (**C**) Maximum specific growth rate (μmax) in supplemented minimal media (red dots), of the 16 strains carrying *HIS3, LEU2, URA3,* and *MET15* plasmids in all combinations obtained from (***Mülleder et al., 2012***), relative to the frequency of the specific metabotype in SeMeCo (light blue). Dashed line indicates average μmax for the eight most and least frequent metabotypes within SeMeCo colonies. (**D**) SeMeCo re-established on minimal media supplemented with uracil. After 7 days of

*Figure 5. continued on next page*

*Figure 5. Continued*

growth (with re-spotting every two days), SeMeCo adapted with an entirely different composition of metabotypes (green box plot) compared to original SeMeCo colony composition (grey bars). Uracil producing cells decline, including the FourP genotype, so that 97% of cells are cooperating auxotrophs.

The following figure supplements are available for Figure 5:

**Figure supplement 1.** Abundance over time of a fluorescent labelled frequent (*HIS3, LEU2, MET15, ura3Δ*) and rare (*his3Δ, leu2Δ, URA3, MET15*) genotype spiked into SeMeCo, as measured by FACS.

Despite its dominant auxotrophic composition, the SeMeCo community could maintain a wild-type like exometabolome, metabolic efficiency, as well as cell viability, implying that this type of cooperation is a robust physiological property. Hence yeast's natural metabolite export and import capacities are wholly sufficient to support co-growth on the basis of metabolite exchange.

The establishment of SeMeCos was not facilitated by mixing the auxotrophs in a higher order combination either. This is consistent with the notion that losing more metabolic genes reduces biochemical capabilities and does not add new ones. The key of the SeMeCo system is instead to allow the progressive self-establishment of the community starting from the single cell (*Figure 2*). Metabolic feedback regulatory systems therefore do not inhibit metabolite export in general but prevent cooperative co-growth when already pre-established co-cultures are mixed (*Müller et al., 2014*; *Shou et al., 2007*). A possible role of these mechanisms could perhaps prevent the spread of foreign, potentially cheating, cells that derive from a competing yeast colony. We could replicate behaviour which is in favour of such an assumption; By spiking into SeMeCo a cell culture possessing the same genotype as a frequent (*HIS3 LEU2 ura3Δ MET15*) and rare (*his3Δ leu2Δ URA3 MET15*) genotype (*Figure 5A*), we observed that both genotypes were rapidly depleted from the pre-established SeMeCo, irrespective of the frequency of the respective genotype in SeMeCo (*Figure 5—figure supplement 1*).

Studying the genotypic composition of SeMeCo implied that there are a defined set of interactions underlying the properties of the cooperating community, which maintained a similar population composition involving eight reproducibly concentrated metabotypes when established independently. This indicates that this quantitatively defined community composition was most effective in metabolic cooperation. This finding may close an important gap in the understanding of the evolution of multicellularity; If a defined composition is most effective in cooperative growth, a selection advantage could be provided by any sort of physical bonding which can maintain cooperation partners in the defined equilibria, and would in addition, provide additional protection against the invasion of cheating cell types. Indeed, the exometabolome data implies that the number of metabolite exchange interactions among co-growing cells could be significant. The colony exometabolome contained a vast array of biomolecules, including the majority of amino acids (*Figure 1C*) (*Castrillo et al., 2007*; *Paczia et al., 2012*; *Silva and Northen, 2015*). The finding that yeast cells prefer uptake over synthesis of amino acids and uracil, even when they are genetically prototrophic, shows that exchange interactions will readily establish once the cellular environment has acquired a critical concentration of metabolites. This has implications for the interpretation of metagenomic studies and cheating/benefactor experiments, as for this reason, it cannot be concluded from the genetic presence or absence of a single metabolic pathway, or from following the synthesis of a single metabolite, how many other metabolites are being exchanged as well.

Even without selective pressure, both wild-type yeasts and SeMeCo established an amino-acid-rich exometabolome on minimal media (*Figure 1C*) and engaged in metabolite uptake when nutrients were available (*Figure 1E*). This implies that these features are a natural property of yeast and raises the question of why natural yeast communities are not composed of co-growing, genetic, auxotrophs. To answer this question, one needs to keep in mind that possessing metabolic genes in the genome is not equal to the pathway being constantly active; Indeed, prototrophs can flexibly switch from self-synthesis to amino acid uptake (*Figure 1*). Being genetically prototrophic hence gives a higher level of metabolic flexibility, as prototrophic cells can re-activate a synthetic pathway when required. Unlike in SeMeCo, the genotype of a cell in a natural community is not in essence reflecting its metabotype or its metabolic role in the community. Additionally, the natural life cycle

of yeast involves the formation of endospores, which are important for enduring starvation, and to spread between habitats. Without a prototrophic genotype, a single spore can no longer establish a colony on its own as genetic auxotrophy would interrupt the yeast life cycle. Second, only a fraction of the natural yeast life cycle occurs under exponential growth that requires abundant carbohydrate and nitrogen supply. The maintenance of a prototrophic genotype both in *S. pombe* and in *S. cerevisiae* wild isolates (*Jeffares et al., 2015*; *Liti et al., 2009*) is hence fully compatible with the presence of elaborate amino acid and nucleotide exchange mechanisms.

The finding that these cells fully shift from self-synthesis to uptake for histidine, leucine, methionine, and uracil, once these metabolites are provided, has direct implications on research using yeast, a primary eukaryotic model organism in genome-scale studies. A majority of yeast genetic experiments are conducted in auxotrophic strains, requiring amino-acid supplemented or rich media compositions. Important parts of biosynthetic metabolism (amino acid biosynthesis can account for up to 50% of metabolic flux towards biomass) may have thus stayed silent in a significant amount of functional genomics experiments. The effects of metabolic–genetic interactions on cellular physiology could thus substantially exceed our current knowledge and could be discovered upon switching to minimal nutrient supplementations. In this context, SeMeCos are simple to handle, establish rapidly and are easy to analyse, and therefore represent an effective and broadly applicable eukaryotic model system to study both cooperativity and effects of metabolism in the laboratory.

In summary, using histidine, leucine, methionine, and uracil as model metabolic pathways for exchangeable metabolites, we found that *S. cerevisiae* cells prefer these nutrients' uptake over their self-synthesis and maintain an amino-acid-rich exometabolome in the extracellular colony space, indicators of ongoing inter-cellular metabolite exchange. Although yeast is known to fail in compensating for auxotrophy in pairwise and higher order co-culture experiments, the cells did successfully enter a state of metabolic cooperative growth upon exploiting stochastic plasmid segregation so that a single cell could progressively develop into a complex heterogeneous community. Composed of auxotrophic cell types that are non-viable on their own, SeMCo communities were able to overcome metabolic deficiencies and maintain metabolite concentrations and robustness similar to wild-type cells. Additionally, cooperation had imposed different metabolic roles on contributing cells. Progressive community formation thus reveals that yeast possesses full capacity to exchange anabolic metabolites at growth relevant quantities and readily establishes a non-cell-autonomous metabolism within complex but defined community structures.

## Materials and methods

### Methods summary

Yeast cells were grown under standard conditions on synthetic minimal (SM or EMM), SC and rich (YPD or YES) media. Plasmid segregation was calculated according to *Christianson et al. (1992)*, by monitoring plasmid retention after cells are shifted from non-selective to selective media, and by expressing the number of cells that have lost the marker as a function of generation time. Metabolites were quantified after quenching using an online UPLC-coupled 6460 (Agilent Technologies, Waldbronn, Germany) triple quadrupole mass spectrometer. Confocal fluorescence microscopy was conducted with a SP5 confocal on a DMI6000 inverted microscope (Leica) using a 10x/0.3 HC PL Fluotar Air objective.

### Yeast strains, plasmids, and growth media

All experiments involved, unless otherwise indicated, used BY4741 yeast strain (*his3Δ1, leu2Δ0, ura3Δ0, met15Δ0*)(*Brachmann et al., 1998*) with prototrophy restored by complementation either with vectors p423 (*HIS3*), pRS425 (*LEU2*), p426 (*URA3*), and pRS411 (*MET15*) (*Christianson et al., 1992*; *Mumberg et al., 1995*; *Sikorski and Hieter, 1989*), with the centromeric vector (minichromosome) pHLUM (*Mülleder et al., 2012*; *Addgene* number: 40276), or with the fluorescent protein vectors yEpCFP_HIS (*HIS3*), yEpSapphire_LEU (*LEU2*), yEpVenus_URA (*URA3*) (*Bilsland et al., 2013*), and pRS411-*GPD*pr-mCherry (*MET15*) (*Table 1*). Cloning was conducted according to standard procedures; oligonucleotides are listed in *Table 2*.

All experiments involving wild-type yeast strains were carried out using as indicated, and using YSBN5, a prototrophic haploid variant of *S. cerevisiae* S288c (*Canelas et al., 2010*). For microscopy

analyses of colony spatial organisation, BY4741 had prototrophy restored by complementation with the above fluorescent protein vectors (*Table 1*). For fluorescence-activated cell sorting (FACS), BY4741 was used with mCherry labelled derivatives of pHLUM, pHLM-*GPD*pr-mCherry, and pUM-*GPD*pr-mCherry (*Table 1*). For *S. pombe* experiments, ED666 yeast strain (ade6-M210 ura4-D18 leu1-32) was used that had uracil and leucine prototrophy restored by complementation with vectors pFS118 (*ura4*$^+$) and pREP41-MCS+ (*LEU2*) (*Table 1*).

Yeast was cultivated if not otherwise indicated at 30°C, in rich (YPD; 1% yeast extract [Bacto], 2% peptone [Bacto] or YES; [Formedium; 35.25 g/L]), complete supplemented synthetic media (SC; CSM complete supplement mixture [MP Biomedicals; 0.56 g/L], YNB, yeast nitrogen base [Sigma; 6.8 g/L]), or minimal supplemented synthetic media (SM; YNB [Sigma; 6.8 g/L] or EMM; [Formedium; 32.3 g/L]), with 2% glucose (Sigma) as the carbon source. Media recipes and amino acid compositions for *S. cerevisiae* were used as previously published (*Mülleder et al., 2012*).

**Table 1.** Strains and plasmids used in this study.

| Name | Description | Reference |
|---|---|---|
| **Strains** | | |
| BY4741 | *MATa, his3Δ1 leu2Δ0 met15Δ0 ura3Δ0* (ATCC 201388) | (*Brachmann et al., 1998*) |
| BY4741 *FLO*$^+$ | Derived from tetrad dissection after crossing and sporulating a flocculating BY4741 strain derived from the knock out collection (*Δtpo1*)with BY4742 and isolating a *FLO + TPO1 wild-type* progeny | This study |
| YSBN5 | *MATa*, FY3 ho::Ble | (*Canelas et al., 2010*) |
| ED666 | *h*$^+$ ade6-M210 ura4-D18 leu1-32 | Bioneer Cat. No. M-3030H |
| **Plasmids** | | |
| p423GPD | 2 μ vector with *HIS3* marker | (*Mumberg et al., 1995*) |
| pRS425 | 2 μ vector with *LEU2* marker | (*Christianson et al., 1992*) |
| p426GPD | 2 μ vector with *URA3* marker | (*Mumberg et al., 1995*) |
| pRS411 | Yeast centromeric vector with *MET15* marker | (*Brachmann et al., 1998*) |
| pHLUM | Yeast centromeric vector with *HIS3, URA3, LEU2* and *MET15* markers (minichromosome). (Addgene number: 40276) | (*Mülleder et al., 2012*) |
| pFS118 | Yeast high-copy vector with endogenous promoter for *ura4*$^+$ (Addgene number: 12378) | (*Sivakumar et al., 2004*) |
| pREP41-MCS+ | Yeast high-copy vector with endogenous promoter for *LEU2*. (Addgene number: 52690) | A gift from Michael Nick Boddy |
| p416GPD | Yeast centromeric vector with endogenous promoter for *URA3* | (*Mumberg et al., 1995*) |
| pHS12-mCherry | Yeast vector with mCherry fluorescent tag and *LEU2* marker. (Addgene number: 25444) | A gift from Benjamin Glick |
| p426-*GPD*pr-mCherry | Yeast 2 μ vector with endogenous promoter for *URA3* marker and a *GPD* promoter for mCherry fluorescent tag | This study. Derived from p416GPD, pHS12-mCherry and p426GPD |
| pRS411-*GPD*pr-mCherry | Yeast 2 μ vector with endogenous promoter for *MET15* marker and a *GPD* promoter for mCherry fluorescent tag | This study. Derived from p426-*GPD*pr-mCherry and pRS411 |
| yEpVenus_URA | Yeast 2 μ vector with *TDH3*-promoter-driven Venus (YFP) and *URA3* marker | (*Bilsland et al., 2013*) |
| yEpCFP_HIS | Yeast 2 μ vector with *TDH3*-promoter-driven CFP and *HIS3* marker | (*Bilsland et al., 2013*) |
| yEpSapphire_LEU | Yeast 2 μ vector with *TDH3*-promoter-driven Sapphire (a UV-excitable GFP) and *LEU2* marker | (*Bilsland et al., 2013*) |
| pHLM-*GPD*pr-mCherry | Yeast centromeric vector with a *GPD* promoter for mCherry fluorescent tag | This study. Derived from pHLUM |
| pUM-*GPD*pr-mCherry | Yeast centromeric vector with a *GPD* promoter for mCherry fluorescent tag | This study. Derived from pHLUM |

**Table 2.** Oligonucleotides used to create expression plasmid p426-*GPD*pr-mCherry.

| Name | Sequence |
| --- | --- |
| mCherry_Bam_Sac_fw | AAGAAGAGCTCAAAAGGATCCGGG**ATG**GTGAGCAAGGGCGAGG |
| mCherry_Xho_rv | CCTTTTCTCGAGCTTGTACAGCTCGTCCATGC |

## Auxotrophy co-cultures

For *S. cerevisiae*, auxotrophic derivatives of prototrophic BY4741 (*Mülleder et al., 2012*) were cultured alone or mixed in combination with other auxotrophs, and 1.1e05 cells of individual or mixed auxotrophs were spotted on respective selective media. Growth was then documented after 2 days incubation at 30°C. For the flocculation experiments, a *FLO⁺* derivative of BY4741 was obtained by back-crossing and tetrad dissection of a *tpo1Δ* (YLL028W) strain obtained from Euroscarf (Frankfurt, Germany). For *S. pombe*, 1.9e04 cells of auxotrophic derivatives of ED666 $h^+$, prototrophic for leucine and uracil, were spotted alone or mixed together on corresponding selective media. Growth was then documented after 2 days incubation at 30°C.

## LC-MS/MS-based quantification of amino acids and uracil

All proteogenic amino acids (except for cysteine) and uracil, citrulline, and ornithine were analysed by selective reaction monitoring (SRM) using an online coupled UPLC (1290 Infinity, (Agilent))/ triple quadrupole mass spectrometer ( 6460, (Agilent)) system. The compounds were separated by hydrophilic interaction chromatography on an ACQUITY UPLC BEH amide column (2.1 mm × 100 mm, 1.7 μm) by gradient elution. Solvent A consisted of 95:5:5 acetonitrile:methanol:water, 10 mM ammonium formate, 0.176% formic acid, and solvent B of 50:50 acetonitrile:water, 10 mM ammonium formate, 0.176% formic acid. The gradient conditions were 0–07 min 85% B, 0.7–27–2.55 min 85–585–5% B, 2.55–255–2.75 5% B, 2.75–275–2.8 min 5–855–85% B and 2.8–38–3.25 min 85% B at a constant flow rate of 0.9 mL/min and 25°C column temperature. SRM (Q1/3 settings) are given in *Table 3*. Metabolite signals were automatically integrated using Masshunter (Agilent) corrected after manual inspection and quantified by external calibration.

## Determination of giant colony intra- and extracellular amino acid concentrations

Cells were spotted on SM solid media and incubated at 30°C in FLUOstar OPTIMA plate reader (BMG LABTECH, Aylesbury, United Kingdom) to establish giant colony. Cells were collected at 26 hr (exponential phase) and re-suspended in $H_2O$. Cells were then gently centrifuged, and pellet (intracellular) and supernatant (extracellular) fractions were separated. Metabolites were extracted from both fractions using 75% boiling ethanol containing l-taurine as an internal standard. Here, samples were left to incubate with extraction solvent in water bath (80°C) for 2 min then mixed vigorously. Incubation and vigorous mixing step was then repeated. Solvent was evaporated using a Concentrator plus Speed Vac (Eppendorf, Hamburg, Germany) and samples were reconstituted in 50 μL 80% ethanol with intracellular fraction diluted 1:5 with 80% ethanol. All samples were submitted to LC-MS/MS and metabolite identification, and quantification was then performed as in '*LC-MS/MS based quantification of amino acids and uracil*'. Data was illustrated following correction to the internal standard of amino acid abundances from both intra- and extracellular fractions.

## Nutrient uptake rates

S. *cerevisiae* strains were transferred from cryo-preserved cultures to SC solid media, grown for 2 days and selected on SM solid media, supplemented only with required amino acids/ nucleobases for 1 day. Pre-cultures were inoculated in 1.5 mL SM containing the minimal supplementation and cultured O/N at 30°C. Main cultures were started at an $OD_{595}$ of 0.15 in 1.5 mL of SC media in deep well 96-well plates and cultured in a Titramax (Heidolph, Schwabach, Germany)for 30 hr (950 rpm, 30°C, 4 mm stirring bead/ well). Samples of 50 μL were harvested every 3 hr, where cells were removed by centrifugation (3000 g, 5 min) and the supernatant diluted 1:20 in absolute ethanol. Then, 1 μL of supernatant was used for quantification of extracellular metabolites by LC-MS/MS.

**Table 3.** SRM transitions for quantification of amino acids and uracil.

| Compound name | Compound abbreviation | SRM transition | Fragmentor (V) | Collision energy (V) | Polarity |
|---|---|---|---|---|---|
| Uracil | U | 111.0 > 42.1 | 62 | 9 | -— |
| Phenylalanine | F | 166.1 > 120 | 100 | 9 | + |
| Leucine | L | 132.1 > 86 | 80 | 8 | + |
| Tryptophan | W | 205.1 > 188 | 85 | 5 | + |
| Isoleucine | I | 132.1 > 86 | 80 | 8 | + |
| Methionine | M | 150.1 > 104 | 40 | 8 | + |
| Taurine | Tau | 126 > 44.1 | 110 | 16 | + |
| Valine | V | 118.1 > 71.9 | 100 | 10 | + |
| Proline | P | 116.1 > 70.1 | 100 | 13 | + |
| Tyrosine | Y | 182 > 165 | 90 | 5 | + |
| Alanine | A | 90 > 44.1 | 50 | 8 | + |
| Threonine | T | 120.1 > 74 | 80 | 9 | + |
| Glycine | G | 76 > 30.1 | 50 | 5 | + |
| Glutamine | Q | 147.1 > 84 | 50 | 16 | + |
| Glutamate | E | 148.1 > 84.1 | 75 | 10 | + |
| Serine | S | 106 > 60 | 40 | 9 | + |
| Asparagine | N | 133.1 > 74 | 80 | 9 | + |
| Aspartate | D | 134.1 > 74 | 80 | 10 | + |
| Histidine | H | 156.1 > 110.2 | 80 | 12 | + |
| Arginine | R | 175.1 > 70 | 100 | 15 | + |
| Lysine | K | 147.1 > 84 | 50 | 16 | + |
| Citrulline | Cit | 176 > 159 | 60 | 4 | + |
| Ornithine | O | 133 > 70 | 90 | 10 | + |

'gcFitModel' function from 'grofit' R package (*Kahm et al, 2010*) was used to estimate the uptake rate of histidine, leucine, uracil, and methionine in different auxotrophic strains.

## Uracil biosynthetic intermediates quantification

After O/N pre-culture S288c *MATa* yeast without auxotrophies and *ura3Δ* yeast (S288c, *MATa*) (*Mülleder et al., 2012*) were grown in 30 mL SM in shake flasks at 30°C, 300 rpm. The media contained either (i) no additives, (ii) uracil (20 mg/L) (iii) or uracil (20 mg/L), leucine (60 mg/L), methionine (20 mg/L) and histidine (20 mg/L). During mid-exponential growth ($OD_{595}$ between 0.7 and 1.2), 1 mL samples of the cultures were quenched in 4 mL -40°C 60% methanol, 10 mM $NH_4$-acetate. After centrifugation (-9°C, 4500 g), the cell pellet was stored at -80°C until extraction.

Prior to extraction, [13]C-yeast internal standard was spiked into the cell pellets, which were subsequently extracted with 1 mL 80°C 75% ethanol, 10 mM $NH_4$-acetate for 3 min. During extraction, the suspension was vortexed on a 0.5–15–1 min time interval. After extraction, the suspension was centrifuged (-9°C, 4500 g) and the supernatant, hence the extract, was dried in a vacuum centrifuge before being stored at -80°C until measurement. For LC-MS measurements, the dried extracts were dissolved in 50–10050–100 μL $H_2O$.

The metabolites were separated with reversed phase ion-pairing chromatography on a Acquity UPLC (Waters, Cheshire, United Kingdom) with a Waters Acquity T3-endcapped column (150 mm, 2.1 mm, 18 μm) as described in (*Buescher et al., 2010*). Subsequently, the metabolites were analysed with a TSQ quantum ultra triple quadrupole mass spectrometer (Thermo Fisher Scientific, Waltham, MA) (*Buescher et al., 2010*). Specifically, the metabolites were ionised with an electro spray (ESI) and the mass spectrometer was run in negative mode with SRM. The SRM transitions used are

described in (*Buescher et al., 2010*) and for orotidine-monophosphate, where no standard was available, we used the phosphate fragment (m/z 367 → 79) trajectory (*Horai et al., 2010*). The obtained data was integrated with an in-house software and normalised to $^{13}$C-internal standard and $OD_{595}$, hence biomass. The median value of different replicates were then scaled and used to illustrate data.

## Determination of plasmid segregation rate

Plasmid stability (segregation) of vectors p423 (*HIS3*), pRS425 (*LEU2*), p426 (*URA3*), and pRS411 (*MET15*) was determined according to *Christianson et al. (1992)*. BY4741 (*his3Δ1, leu2Δ0, ura3Δ0, met15Δ0*) either transformed with one or all four plasmids, respectively, for either the four non-fluorescent or fluorescent vectors, were cultured in 25 mL of YPD media for 48 hr then plated at 1:100,000 dilution on YPD solid media. Plasmid retention was then calculated by replica plating CFUs from YPD solid media onto selective solid media. Number of doublings (g) and segregation rate (m) were calculated as in (*Christianson et al., 1992*).

## Calculation of colony compositions based on segregation rate

Segregation rates of p423 (*HIS3*), pRS425 (*LEU2*), p426 (*URA3*), and pRS411 (*MET15*) were simulated over generation time in R by iterative cycling (looping). The script is given in *Supplementary file 1*. Plasmid abundances were binned by plasmid number (0 to 4) and illustrated with R package 'ggplot2' in terms of auxotrophy.

## Growth analysis

Unless otherwise indicated, cells were first spotted and grown for 2 nights on SM solid media to establish a giant colony. The colony was then re-suspended in $H_2O$ and diluted to 3.4e03 cells in 200 μL SM, and their optical density ($OD_{595}$) was recorded in a FLUOstar OPTIMA plate reader (BMG LABTECH) every 20 min for 40 hr at 30°C. Both maximum specific growth rate (μmax) and lag phase were determined from growth curves using a model-richards fit from the R 'grofit' package (*Kahm et al, 2010*). For determining dry biomass, colony was re-suspended in $H_2O$ and normalised to 1.1e07 cells in 100 mL SM, then incubated for 72 hr at 30°C and pelleted. Pellets were dried for 5 days at 50°C and then weighed to obtain dry biomass. μmax for individual metabotypes was determined in batch SM culture (50 mL) and supplemented accordingly for the different auxotrophic requirements.

## Cell viability of individual cells in SeMeCo colonies

Cells from giant colonies of SeMeCo and YSBN5 were grown to exponential growth phase in SC media, and cell number was then measured with a CASY Model TTC (Roche Innovatis, Switzerland) cell counter. Cells were then diluted and plated on solid SC media to establish individual CFUs and the number of CFUs with initial cell number were compared.

## Spatial organisation of colony via fluorescence microscopy

A micro-colony of BY4741 with prototrophy restored by complementation with the fluorescent protein vectors yEpCFP_HIS (*HIS3*), yEpSapphire_LEU (*LEU2*), yEpVenus_URA (*URA3*) (*Bilsland et al., 2013*), and pRS411-*GPD*pr-mCherry (*MET15*) (*Table 1*) was grown for 2 nights on SM. Prior to imaging, colony was embedded in 2% agarose (Type I-B; Sigma) and gently transferred to a μ-slide glass bottom (ibidi). Cells were imaged with a DMI6000 inverted Leica SP5 confocal microscope, using a 10×/0.3 HC PL Fluotar Air objective, running LAS AF software (version 2.7.3.9723). Fluorescence for each marker was separated by excitation (CFP: 458 nm, Sapphire: 405 nm, Venus: 514 nm and mCherry: 561 nm). In our hands, the Sapphire-*LEU2* was also visible under the imaging conditions used to visualise the Venus-*URA3*. For this reason, we removed Venus-*URA3* channel from the colony image. A look-up table was applied to each channel post-acquisition to allow visualisation of the different channels together using ImageJ software.

## Calculation of distances between auxotrophs and their corresponding prototrophs

For the microscopy image showing fluorescent-labelled cells in a colony, prototrophs and auxotrophs were identified as being present or absent using several cut-offs for fluorescent signal intensity. Based

on these different cut-off values (0.1, 0.2, 0.3, and 0.4), the monochrome fluorescence microscopy images of each individual marker (HIS3, LEU2 and MET15) were recognised as metabolite producing (prototrophic) or requiring (auxotrophic) pixels. The auxotrophic pixels were marked black, prototrophic pixels white, and the background was illustrated grey to separate it from the colony pixels. For each auxotrophic pixel, the distance to the next prototrophic pixel was calculated and distance values for the minimum, mean, overall maximum, and maximum for 90% of cells were taken for each marker.

### Fluorescence-activated cell sorting of labelled frequent and rare metabotypes

Genotypes depicting SeMeCo frequent (HIS3, LEU2, MET15, ura3Δ) and rare (his3Δ, leu2Δ, URA3, MET15) metabotypes were reconstructed by transforming mCherry labelled derivatives of pHLUM (pHLM-GPDpr-mCherry and pUM-GPDpr-mCherry, respectively) into BY4741. Strains were spiked into established SeMeCo in 1:10 (frequent or rare metabotype: SeMeCo) ratio taken from their established giant colonies on selective media (where plasmid segregation is ongoing). Abundance of fluorescent cells was monitored immediately after mixing and approximately 48 hr after re-establishment of giant colony on minimal solid media with a BD LSRFortessa cell analyser. Data analysis was performed with FlowJo.

## Acknowledgements

We thank Uwe Sauer (ETH Zurich) for support in metabolite measurements and scientific discussion and Elizabeth Bilsland for kindly donating the fluorescent protein plasmids to help determine colony spatial organisation.

## Additional information

### Funding

| Funder | Grant reference number | Author |
| --- | --- | --- |
| Wellcome Trust | RG 093735/Z/10/Z | Markus Ralser |
| European Research Council | StG 260809 | Markus Ralser |
| Isaac Newton Trust | RG 68998 | Markus Ralser |
| Austrian Science Fund | J3341 | Markus A Keller |

The funders had no role in study design, data collection and interpretation, or the decision to submit the work for publication.

### Author contributions

KC, JV, MM, SM, NL, EC, LMF, MTA, SC, MAK, Acquisition of data, Analysis and interpretation of data, Drafting or revising the article; MR, Conception and design, Analysis and interpretation of data, Drafting or revising the article

### Author ORCIDs

Markus Ralser, http://orcid.org/0000-0001-9535-7413

## Additional files

### Supplementary files

• Supplementary file 1. Script used for the simulation of plasmid segregation over time, using R (r-project.org).

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
