## [Decision Letter]

Thank you for submitting your work entitled "Cell-cell heterogeneity emerges as a consequence of metabolism in a cooperating eukaryotic cell community" for peer review at *eLife*. Your submission has been favorably evaluated by Diethard Tautz (Senior Editor) and four reviewers, one of whom, Mohan Balasubramanian, is a member of our Board of Reviewing Editors.

The reviewers have consulted with one another to come to some understanding of what revisions should be required to produce a revised manuscript suitable for publication in *eLife*. The reviewers raise a general concern about using your approach to delve into aspects of yeast cell biology and physiology and we are concerned that this could require more than the two months we normally allow for return of a revised manuscript.

The referees points are provided verbatim, however, please pay attention to and be sure to address the key points in this letter.

Essential revisions:

The referees liked the new approach using plasmid loss to generate a synthetic yeast community exchanging nutrients, but questioned how the work may provide information on yeast physiology. This could be achieved, for example, using the *Sigma1278b* strain strategy mentioned by Reviewer #2.

There were also concerns about the lack of cell biological analyses and I would encourage you to address these satisfactorily with experiments and discussion.

Please pay close attention to the point raised by Reviewer #3 on the exo versus endometabolome differences in the context of the colony and address it satisfactorily with experiments and discussion.

Reviewer #1:

This is in interesting manuscript from Ralser and colleagues using a new approach to generating complex yeast community that exchange metabolites, and such exchange is essential for colony viability. The authors show that they can generate synthetic colonies that contain auxotrophs for 4 different amino acids/nucleobases. At one level this is exciting, but at another level, I am unclear what we have learned about yeast physiology from this work.

Major concerns:

1) The plasmid loss based approach to generating the SeMeCo colony is a clever trick, but how does it generate the SeMeCo. Delving into the mechanism of how gradual loss of plasmids in a mixed population (as opposed to mixing various auxotrophs) leads to SeMeCo will be a great strength.

2) The paper lacks a cell biological analysis of the phenomenon. For example, Muller, Murray use a fluorescence based assay to look at the position of the two genotypes in the colony. Admittedly, the situation here is more complex with a larger number of genotypes, but is still not insurmountable and some cell biological insight into colony organization will help.

3) It is also not strictly correct to say that exchange of metabolites/cooperation does not happen in yeast, since the papers by Shou and Murray labs (cited in this paper as well) do generate a synthetic cross-feeding colony, except that these authors had to down-regulate feedback mechanisms involved in amino acid metabolism. What the current paper has done is to increase the complexity and this has been achieved using a different clever approach. This needs to be better articulated.

Reviewer #2:

Campbell and coworkers report the establishment of a community of yeast cells with different auxotrophies that manages to grow through exchanges of metabolites. Specifically, they started from an auxotrophic mutant that was restored to prototrophy through transformation with plasmids that complemented the missing biosynthetic genes. As this prototroph divides, some of the plasmids are lost in some of the progeny, resulting in a complex but reproducible community of prototrophs and several auxotrophs; demonstrating that the cells must be exchanging nutrients that support growth of the different auxotrophs.

The main conclusion from this elegant work is that (eukaryotic) yeast cells are able to exchange metabolites. I am quite excited about this result, because it is getting at an underexplored, yet very interesting and possibly important aspect of cooperation, community formation and multicellularity in eukaryotes. As such, I support publication of this paper in *eLife*. However, I also have several important questions and remarks.

Major concerns:

1) I think that the main story of the paper is a bit underdeveloped still, and that the second part of the paper (where the authors measure stress resistance, etc.) is perhaps less essential and a bit unconnected from the main story. To me, the most important question is how important and realistic the findings are for natural conditions. This particular experiment and the establishment of SeMeCos represents a scenario that is not very realistic in nature, where all the genes are maintained more or less stably in the genome instead of residing on unstable plasmids. The authors find that auxotrophic mutants that are mixed together do not seem able to develop some sort of mutualistic community. Moreover, in natural conditions, exchange of metabolites between yeast cells may not be very relevant, since if it were, one would expect that feral yeast colonies or communities would contain a very high rate of (different) auxotrophs? This also makes me wonder whether SeMeCo's develop mostly because of the cost of maintaining the plasmids, rather than a benefit of losing specific metabolic pathways to specialize in others? Last but not least, I also wonder whether the lab yeast strain BY4741, a derivative of S288C, which is known to be impaired in several mechanisms related to community formation (biofilms, pseudohyphal growth, mats…), is a wise choice for these experiments.

Some additional experiments may tell us much more about the real-life importance of metabolite exchange in yeasts. Firstly, I would argue to repeat the key experiments in a strain that is capable of forming communities, like for example *Sigma1278b*. At first sight, this might seem like a prohibitive amount of work, but it is not since it simply entails using some of the available auxotrophs (or generating a few new ones through transformation) to see if they can support each other in liquid suspension, colonies or biofilms/mats. If so, this would mute a lot of doubts (including whether the cost of maintaining a plasmid is a player). Second, I would also suggest transforming Sigma with (at least two of) the plasmids to investigate if and how this strain realizes SeMeCo's similar to S288c. If the authors decide to follow this suggestion, I would also argue for the inclusion of constitutively expressed fluorescent markers, so that they can easily count the different lineages through cytometry and also follow their spatial distributions in colonies/biofilms (see second remark below).

Last but not least, I think that the authors need to state more clearly what they think is the relevance of their findings in natural conditions – they seem to avoid this question a bit instead of being very explicit about it and discussing it in great detail. One very interesting bit of information would be if and how many auxotrophs are recovered from nature? Is there any data available? Or can the authors explore this themselves?

2) One of the most intriguing questions is whether the different auxotropic mutants arrange into some kind of spatial pattern in the colonies, so that complimentary strains (that can compensate each other's auxotrophies) would be in close proximity. This should be relatively easy to study by integrating a fluorescent marker in each of the plasmids (e.g. three dyes with a very different spectrum, like CFP, YFP and RFP).

3) It is unclear to me whether it is sufficient to keep cells in exponential phase to exclude the influence of dying cells (which could contribute nutrients, rather than through exchange between living cells). The fact that the "exometabolome" differs in composition from that inside of living cells may not be the ultimate proof, as it seems possible that dead cells do not only contribute metabolites that were free molecules when the cell was still alive, but also through (active or passive) breakdown of biomolecules after the cells disintegrate. Perhaps one way to tackle this is to let cultures grow well into stationary phase and/or add a certain number of dead cells and monitor if and how this changes the dynamics. Another way would be to investigate colonies, as it is known that cells in the center of colonies experience starvation and cell death (see for example work by Palkova and colleagues).

4) The SeMeCo's seem to show convergence towards a stable state (with a defined and complex ratio of different auxotrophic mutants). Are we sure that this is indeed a stable state? And why do not all prototrophs disappear? Would it be worth letting some communities grow much longer to see what happens in the long term?

Reviewer #3:

Cells within community may cooperate by exchanging metabolites, easing the metabolic burden on individuals. Budding yeast "wild type" auxotrophs do not complement metabolic deficiencies when grown in co-cultures, leading to speculations that yeast may be deficient in some aspect of metabolite export. The manuscript under review describes budding yeast communities capable of metabolic exchange and cooperation. The key trick is to allow those communities to self-establish from a single cell initially auxotrophic for synthesis of three amino acids and uracil in which auxotrophy is covered by complementing plasmids. Each of these plasmids can be lost stochastically during cell division, leading to a mixture of possible phenotypes. The authors show that such stochastic plasmid loss leads to the emergence of heterogeneous colonies where auxotrophic neighbors grow and exchange metabolites between each other. It is a clever approach leading to a neat observation and the manuscript describes a lot of experimental work. My biggest problem with this story is that there is no attempt to get at the underlying mechanism and to understand just how SeMeCos differ from cells in co-culture.

Major concerns:

Are individual SeMeCo cells sensitive to end-product feedback? Could they have accumulated mutations inactivating the feedback inhibition? Other mutations/genome rearrangements? Do cells maintain ploidy?

Somewhat related, is it possible to reproduce mutualistic growth from scratch, in co-cultures, using mixed populations of cells with metabotype composition similar to SeMeCos? Perhaps in conditioned SeMeCo media, since the authors suggest that "exchange interactions will readily establish once the cellular environment has acquired a critical concentration of metabolites".

I do not quite understand why SeMeCos transferred to the liquid minimal medium – and at very low dilutions (Figure 3) – do not show longer lag phase as compared to their parent strain and grow normally. Do they overproduce and/or over-secrete metabolites under these conditions? Would it be expected that metabolite concentration is initially very low? Or are those initially proliferating cells prototrophs that may later start losing plasmids, re-establishing the community?

I do not get the logic behind experiments shown in Figure 1. The authors first show that many metabolites are found in extracellular space within yeast colonies. They then argue that these metabolites must be exported rather than released from lysing cells because endo- and exometabolomes of exponentially growing cells are different. I appreciate that growing cells may well export metabolites but why is this detour to exponentially growing cultures where cell lysis is negligible? How is it related to colonies where growth is restricted to a small outward layer and cell viability within a colony center is likely decreased?

Similarly, what is the reason for including an experiment shown in 2Av? The fact that cells lose plasmids in the absence of selection is not particularly surprising.

What is the degree of cell death during community establishment? Somewhat related, what is the efficiency of SeMeCo establishment? Do all cells eventually give rise to a mutualistic community?

Reviewer #4:

This is an interesting paper describing how it is possible to evolve yeast communities where certain members produce specific amino acids/nucleotides and other members consume these metabolites. The authors demonstrate that this community behaves similar to wild type prototrophic strains, both in terms of specific growth rate and in response to stress. The authors present an impressive number of controls and perform a detailed analysis of the established community. So overall this is a sound paper and I do not really have any specific comments to the Results and Discussion. However, I am a little puzzled about the rationale of this work. It is not clear what we are learning in terms of new biology from the study. It is an interesting observation that both phototrophs and auxotrophs take up metabolites at the same rate, but I think this has been observed before. The authors can probably easily revise their paper to take this point into account, as I am certain that they in fact could argue for the rationale of this study.

[Editors' note: further revisions were requested prior to acceptance, as described below.]

Thank you for resubmitting your work entitled "Self-establishing communities enable metabolic cooperation in a eukaryote" for further consideration at *eLife*. Your revised article has been favorably evaluated by Diethard Tautz (Senior Editor), Mohan Balasubramanian (Reviewing editor), and three reviewers. The manuscript has been improved but there are some remaining issues that need to be addressed before acceptance, as outlined below:

As you will see, the referees have asked you to consider and rewrite some points mentioned below and also discuss some caveats that they have noted.

Reviewer #2:

Overall, I think that this paper has matured significantly and should be published. I feel that the new experiments make the study much stronger, and I specifically appreciate the authors' efforts to re-write the Discussion section to better highlight the biological relevance of their findings, as well as the remaining questions.

That said, as I was re-reading the paper, I again started wondering why auxotrophic mutants, even when mixed at the same ratios as found in SeMeCos, fail to establish a growing population; whereas prototrophic cells that contain unstable plasmids are able to form a community with some fractions of the population showing plasmid loss that results in auxotrophy. I understand the authors' argument that feral yeasts often do not show auxotrophies because they may only rarely grow exponentially and because they may go through sexual cycles where they need to be able to survive and divide as single haploid cells. But why would mixing auxotrophs in rich medium not yield the same result as a population where the haploids are gradually formed through plasmid loss as the population grows?

One important potential artefact would be that after plasmid loss, the proteins encoded by the lost plasmids remain present and active in the cells for a few generations. It is well known that protein carry-over can persist for more than 5 doublings, so in principle, it could be possible that the population contains "zombies" – cells that are able to survive and even divide for a few rounds, but whose lineage is eventually bound to disappear when the limiting proteins become too diluted because of divisions and protein turnover… This could also explain why at least a fraction of the population remains prototrophic.

I am sure that the authors have also thought about this possibility and will be able to provide arguments against my speculation. It might be useful to also include these in the paper. One possibility would be to closely examine the fluorescence profiles in the SeMeCos – are there any signs of auxotrophs still showing (reduced) fluorescence, or is there a clear bimodal distribution?

Reviewer #3 [abridged]:

I am fine with the revision – it is a considerably improved and focused version of the manuscript. There are several instances where the text will benefit from further edits. I will point out a few:

The authors mention that they "re-isolated individual prototrophs from the established colonies…". I am somewhat confused – is this a typo and it should be "individual auxotrophs", since the authors attempt and fail to co-culture them later? Or do the authors mean that culturing prototrophs for 48 hour in supplemented medium (liquid? solid?) may induces plasmid loss? Either way, it could be worth clarifying this point.

In the subheading “SeMeCos reveal composition dynamics in response to nutritional changes”, the authors state that: "These findings are hence consistent with the notion that metabolic cooperation prevents population intermixing, so that cooperation partners within the colony retain close proximity to prevent diffusion of the exchanged metabolite". Yet, the authors also show that metabolite exchange is not constrained by colonial growth and does continue in liquid cultures. I wonder if the authors could add a sentence or two to discuss this point.

Reviewer #4:

I have no further comments to this paper. I think the revised version provides a better rationale for the work, and I also think the authors have addressed all the other reviewers’ comments satisfactorily.

---

## [Author Response]

*Essential revisions:The referees liked the new approach using plasmid loss to generate a synthetic yeast community exchanging nutrients, but questioned how the work may provide information on yeast physiology. This could be achieved, for example, using the* Sigma1278b *strain strategy mentioned by Reviewer #2.*

We agree with referee 2 that BY4741, as a S288c derivative strain, has a long laboratory history that possesses (like other lab strains do as well) some metabolic defects. We therefore agree it can be interpreted as a subject of concern that all results shown were obtained in this background. However, only a single gene mutation (*FLO1*) prevents the assembly of biofilm-like structures in S228c (flocculation), so metabolically, the strain is completely capable of forming these types of communities (Smukalla et al., 2008). Although *Sigma1278b* is capable of biofilm formation in a growth-media dependent manner, it remains a very close relative to S288c, with a largely identical genome, similar metabolic capacities, and a long lab history on its own (http://wiki.yeastgenome.org/index.php/History_of_Sigma). We therefore chose to replicate the key experiments in a totally different yeast species, as these experiments not only exclude that the findings are specific to BY4741, but also indicate evolutionary conservation. In addition, we studied a flocculating derivative of BY4741, to exclude that ability of physical attachment influences the experiments outcome.

We chose *S. pombe* as the ideal candidate, being ~400 Myr evolutionary apart from *S. cerevisiae*, and additionally, as *S. pombe* offers a comparable set of laboratory techniques. We performed co-culture experiments using representative *S. pombe* auxotrophs and find that they, like *S. cerevisiae* cells, are also not able to form a co-growing community upon co-culturing (Figure 1).

Next, we also replicated a segregation experiment in the same *S. pombe* strain as Figure 1. Also in *S. pombe*, segregation allowed the cells to establish a cooperating community, as in *S. cerevisiae* (Figure 2—figure supplement 2).

In addition, to addressing whether this phenotype is the impacted by the biofilm forming capacities referred by Reviewer #2, we performed the co-culture experiments also in *FLO*^+^*S. cerevisiae* strain. The ability of *FLO*+ cells to physical attach to one another did not influence the outcome of the experiments. Complementary co-cultures of *FLO*+ auxotrophs did not co-grow in the absence of supplementation (Figure 1—figure supplement 1).

The most important implication for yeast physiology is that, our results being correct, cells within colonies continuously and extensively exchange anabolic metabolites among each other and do not, as the current assumption says, produce metabolites predominantly for themselves. We draw these conclusions from (i) finding metabolites to be highly concentrated between cells, (ii) demonstrating that cells take these up preferentially when they are available and (iii), showing that cells have a highly robust capacity for growing on the basis of this metabolite exchange.

We have extended the discussion about the following implications: that knowledge on ongoing metabolite exchange is essential for understanding the physiology of cells, as it shows that in a community, metabolic networks operate non-autonomously. For instance, this knowledge is essential to interpret the recent results by Shou et al and Murray et al, that mix yeast cells under the assumption that only feedback-modified cells exchange intermediate metabolites, whereas non-modified prototrophs do not (Momeni et al., 2013a, 2013b; Müller et al., 2014; Shou et al., 2007). These results are also essential for interpreting and improving results from metabolic modelling approaches such as flux balance analysis, that intuitively assume cell-autonomous functionality of intermediary metabolism and, at present, do not include transport reactions or assume regular metabolite exchange between cells. For biotechnology, the potential is huge as well, as SeMeCos offer a simple system for establishing cooperative communities; these could be very efficient for the production of cell-toxic substances, as the burden can be shared between cells.

Please also see the individual response to Reviewer #2 that is interested in the yeast ecological implications of the metabolite exchange.

There were also concerns about the lack of cell biological analyses and I would encourage you to address these satisfactorily with experiments and discussion.

We have replicated a series of experiments in this respect as requested by the reviewers. We have introduced plasmids containing fluorescent proteins coupled to auxotrophic markers and did reconstruct SeMeCo and visualise structural organisation of a micro-colony via confocal fluorescence microscopy. These experiments add appreciable visual evidence that metabolic heterogeneity in a SeMeCo colony is substantial (Figure 4).

We next performed advanced image analyses from these microscopy images and show that the distance a metabolite needs to cross to reach a consuming auxotroph from a supplying prototroph is minimal (less than two yeast cell diameters) (Figure 4). This shows that the community establishes a structure due to cooperation.

Please pay close attention to the point raised by Reviewer #3 on the exo versus endometabolome differences in the context of the colony and address it satisfactorily with experiments and discussion.

We have conducted several experimental controls to address the comments of Reviewer #3. First, we have re-isolated different auxotrophic cells from an established SeMeCo to test whether they, after metabolic reset by growing in supplemented media, behave like the original strains and fail to establish cooperation upon co-culturing. As illustrated, they behave like the original strains and fail to grow together upon co-culturing. This excludes that feedback regulatory systems could have acquired mutations while SeMeCos established, or being affected by other genetic alterations (Figure 2—figure supplement 3).

As another requested control by the reviewer, we co-cultured the auxotrophic strains in the composition as quantified in SeMeCo, and in a 1:1:1:1 ratio. As expected, a similar result as in the pairwise co-cultures is obtained. The result is also robust when co-culturing overnight in rich media before spotting to minimal media (Figure 2—figure supplement 4). We would like to note that these controls were, in another place, already part of the first submission, and apologize that they were not highlighted as such.

We also tested whether cell death in SeMeCo is different to that in wild-type cells. Both wild-type cells and SeMeCos did produce the same number of CFU's on complete media. Moreover, we counted the cell numbers with a CASY cell counter before spotting, and compared it with the number of colonies obtained. The numbers are close to a 1:1 ratio, confirming that cell death in exponentially growing wild-type and SeMeCo cells is marginal, and that virtually every cell can give rise to a new colony (Figure 3). We would like to note that the Palková experiments that have been referred to (Váchová et al., 2012), are conducted in old, stationary colonies that establish over several days to a week, where cell death is expected to commence in the regions of the colony where there is no cell growth and the population ageing. In difference, all our experiments are conducted in continuously growing colonies and cultures, were old cells are constantly diluted and never reach a significant percentage of the community.

We also revised our point about the exo vs endometabolome and apologize that it created an unintentional confusion. It is not a problem that a low number of dying cells may contribute to the exometabolome – even in exponential cell growth a (very) low number of cell death might occur and this is a physiological situation in a yeast colony and cannot be prevented. There would also be no way to distinguish this by mass spectrometry or another analytical technique (physically it is the same metabolite, differential labelling is no option as it is not predictive which cell will die first), subsequently, we can only work indirectly, therefore we only use growth conditions where cell death is negligibly low.

Why is this point important at all? We have to exclude that nutrient leakage from dying cells is the main explanation for auxotrophic cell growth in our system. Nutrient recycling from dying cells can play a significant role in chronological ageing experiments, where cells are kept for days or weeks in stationary growth phase and cell death becomes substantial (i.e., studied by Palková and Longo labs). The phenotype, 'adaptive re-growth', can subsequently arise in these stationary ageing yeast cultures as a result of cell lysis. In stationary bacterial cultures, a similar phenotype also occurs, known as 'growth advantage in stationary phase' (GASP). In these stationary cultures, cell growth that is explained through recycling from dying cells, but the novo biomass is not formed. This is the clear difference between these scenarios and our experimental set-up: In our case we can say for certain that de novo synthesis explains cell growth, as we work under exponential growth conditions, where biomass and with it, the total amount of histidine, leucine, methionine, and uracil, doubles with every cell division. Recycling of nutrients from dying cells could – at best – maintain biomass, but never double it exponentially; so on minimal media, all new gained biomass is for certain explained by de novo synthesis. Together with detecting no relevant amount of cell death as expected for an exponential culture Figure 8, we conclude that nutrient release through cell death is overall not relevant for the growth of SeMeCos.

Reviewer #1:*This is in interesting manuscript from Ralser and colleagues using a new approach to generating complex yeast community that exchange metabolites, and such exchange is essential for colony viability. The authors show that they can generate synthetic colonies that contain auxotrophs for 4 different amino acids/nucleobases. At one level this is exciting, but at another level, I am unclear what we have learned about yeast physiology from this work.Major concerns:1) The plasmid loss based approach to generating the SeMeCo colony is a clever trick, but how does it generate the SeMeCo. Delving into the mechanism of how gradual loss of plasmids in a mixed population (as opposed to mixing various auxotrophs) leads to SeMeCo will be a great strength.*

We agree with the line of thinking of the reviewer, but believe the main question the reviewer asks is what prevents the formation of co- growth in the co-culture experiments. The growth of SeMeCo instead, as addressed in this paper, is clearly enabled though metabolite exchange of histidine, leucine, uracil, and methionine, as these are the only limiting metabolites for the auxotrophs to grow. We suspect that feedback control mechanisms might be responsible; but we have no clear idea of what activates them specifically upon co-culturing. We find this problem as interesting as the reviewer, however, its answer will become a complicated study, and will have to be addressed on its own in the future. We are currently applying for funding to extend this work.

2) The paper lacks a cell biological analysis of the phenomenon. For example, Muller, Murray use a fluorescence based assay to look at the position of the two genotypes in the colony. Admittedly, the situation here is more complex with a larger number of genotypes, but is still not insurmountable and some cell biological insight into colony organization will help

We have addressed this experimentally, please see Essential Revision Point #2 above. These new experiments form a new figure (Figure 4). We had originally restrained from the use of fluorescent markers as they might have introduced an additional cost though their synthesis in the establishment of a SeMeCo. However, what we can tell so far yeast seem to tolerate expression of the different GFP derivatives without changing the cell growth properties notably.

3) It is also not strictly correct to say that exchange of metabolites/cooperation does not happen in yeast, since the papers by Shou and Murray labs (cited in this paper as well) do generate a synthetic cross-feeding colony, except that these authors had to down-regulate feedback mechanisms involved in amino acid metabolism. What the current paper has done is to increase the complexity and this has been achieved using a different clever approach. This needs to be better articulated.

We apologize if our quoting of their work was misleading. We have carefully revised the paper.

Reviewer #2:

Major concerns:

*1) I think that the main story of the paper is a bit underdeveloped still, and that the second part of the paper (where the authors measure stress resistanc, etc.) is perhaps less essential and a bit unconnected from the main story. To me, the most important question is how important and realistic the findings are for natural conditions. This particular experiment and the establishment of SeMeCos represents a scenario that is not very realistic in nature, where all the genes are maintained more or less stably in the genome instead of residing on unstable plasmids. The authors find that auxotrophic mutants that are mixed together do not seem able to develop some sort of mutualistic community. Moreover, in natural conditions, exchange of metabolites between yeast cells may not be very relevant, since if it were, one would expect that feral yeast colonies or communities would contain a very high rate of (different) auxotrophs? This also makes me wonder whether SeMeCo's develop mostly because of the cost of maintaining the plasmids, rather than a benefit of losing specific metabolic pathways to specialize in others?*

The reviewer addresses the ecological implications of our study. In response to the general comment, we would like to bring up that there are two points here, and we think both are important for the natural live of *S. cerevisiae*. The questions a) is whether native yeast cells would exchange intermediary metabolites and possess the capacity to do that as described in our work, and b) whether this exchange would result in a situation, were losing biosynthetic genes provide an additional benefit to that, by forming communities were co-growth of different yeast genotypes becomes obligate.

To point a, our results give clear indication that the maintenance of an amino acid rich exometabolome establishes within yeast colonies as a normal property (Figure 1) and the cellular uptake of these metabolites to support growth is a native yeast property as well (Figure 1): The experiments were conducted with prototrophic yeast, grown on minimal media, were no pressure was given to maintain the exometabolome or to uptake the metabolites. Also we see throughout the paper that SeMeCos maintain growth and many physiological parameters that mimic that of colonies composed of genetically prototrophic cells.

The experiments in Figure 1 demonstrate that prototrophic yeast cells can flexibly switch between uptake and self-synthesis of histidine, leucine, uracil and methionine. The critical question for point B), is hence if cells would gain additional advantage by losing this flexibility, and enter a situation were co-growth of different genotypes becomes an obligate situation. One needs to see the answer to this question in the context of the natural life cycle of yeast. This involves a sexual cycle and the formation of endospora, important to endure long periods of starvation, and to spread between habitats. If a yeast cell would lose a prototrophic genotype, a single spora cannot any longer establish a colony on its own. Loss of genetic prototrophy would thus negatively affect the natural life cycle of yeast, with deleterious consequences to its survival, spreading between environments, and evolvability (interruption of genetic recombination, etc.). So even if exchanging metabolites is advantageous when cell grow in a community, there are good reasons for keeping the genes in the genome that allow metabolic flexibility when required and for completing the sexual cycle. The existence and activity of metabolite exchange strategies coupled with the ability to turn biosynthetic pathways on and off, is thus fully compatible with maintaining a prototrophic genome. Perhaps, higher organisms are a good example for that. As humans we obtain all amino acids through our food, yet we maintain biosynthetic pathways for eleven of them, so that we can flexibly activate these pathways when our diet or age requires that. For yeast, the majority (but not all) natural isolates keep genetic prototrophy, both in *S. pombe* and in *S. cerevisiae* (Jeffares et al., 2015; Liti et al., 2009). We have expanded the discussion about this point.

Last but not least, I also wonder whether the lab yeast strain BY4741, a derivative of S288C, which is known to be impaired in several mechanisms related to community formation (biofilms, pseudohyphal growth, mats…), is a wise choice for these experiments.

*Some additional experiments may tell us much more about the real-life importance of metabolite exchange in yeasts. Firstly, I would argue to repeat the key experiments in a strain that is capable of forming communities, like for example* Sigma1278b*. At first sight, this might seem like a prohibitive amount of work, but it is not since it simply entails using some of the available auxotrophs (or generating a few new ones through transformation) to see if they can support each other in liquid suspension, colonies or biofilms/mats. If so, this would mute a lot of doubts (including whether the cost of maintaining a plasmid is a player). Second, I would also suggest transforming Sigma with (at least two of) the plasmids to investigate if and how this strain realizes SeMeCo's similar to S288c.*

We have followed the suggestion of the reviewer, and have replicated the key experiments in (a) *S. pombe*, that is evolutionary distant to *S. cerevisiae*, and (b) in a *FLO*^+^*S228c S. cerevisiae* strain, to exclude that the findings are only relevant for S288c. Please see Essential Revision Point #1 above.

If the authors decide to follow this suggestion, I would also argue for the inclusion of constitutively expressed fluorescent markers, so that they can easily count the different lineages through cytometry and also follow their spatial distributions in colonies/biofilms (see second remark below).

We have now included experiments with fluorescent markers. Please see Essential Revision Point #2.

Last but not least, I think that the authors need to state more clearly what they think is the relevance of their findings in natural conditions – they seem to avoid this question a bit instead of being very explicit about it and discussing it in great detail. One very interesting bit of information would be if and how many auxotrophs are recovered from nature? Is there any data available? Or can the authors explore this themselves?

The answer to this has been included with point #1 above about the ecology.*2) One of the most intriguing questions is whether the different auxotropic mutants arrange into some kind of spatial pattern in the colonies, so that complimentary strains (that can compensate each other's auxotrophies) would be in close proximity. This should be relatively easy to study by integrating a fluorescent marker in each of the plasmids (e.g. three dyes with a very different spectrum, like CFP, YFP and RFP).*

See Essential Revision Point #2. These experiments have now been included. Complementary auxotrophs stay in close proximity. This does not, however, prevent macroscopic differences to establish.

3) It is unclear to me whether it is sufficient to keep cells in exponential phase to exclude the influence of dying cells (which could contribute nutrients, rather than through exchange between living cells). The fact that the "exometabolome" differs in composition from that inside of living cells may not be the ultimate proof, as it seems possible that dead cells do not only contribute metabolites that were free molecules when the cell was still alive, but also through (active or passive) breakdown of biomolecules after the cells disintegrate. Perhaps one way to tackle this is to let cultures grow well into stationary phase and/or add a certain number of dead cells and monitor if and how this changes the dynamics. Another way would be to investigate colonies, as it is known that cells in the center of colonies experience starvation and cell death (see for example work by Palkova and colleagues).

Please see Essential Revision Point #3, where these points have been addressed.*4) The SeMeCo's seem to show convergence towards a stable state (with a defined and complex ratio of different auxotrophic mutants). Are we sure that this is indeed a stable state? And why do not all prototrophs disappear? Would it be worth letting some communities grow much longer to see what happens in the long term?*

We have conducted a stability experiment for a SeMeCo community over a period of ~100 generations. As expected for a living ecological system, it is not 100% identical over this long period, but the variation is in the range not larger ~ 10% and restabilizes (Figure 6).

Author response image 1.****Stability of SeMeCo colony over time.Starting from a SeMeCo micro-colony on minimal media, a giant colony was established and composition was followed by replica plating for 90 generations. Biomass gain is calculated starting from the single cell.**DOI:**
http://dx.doi.org/10.7554/eLife.09943.024

Reviewer #3:

Major concerns:

Are individual SeMeCo cells sensitive to end-product feedback? Could they have accumulated mutations inactivating the feedback inhibition? Other mutations/genome rearrangements? Do cells maintain ploidy?

We thank the reviewer for this suggestion, it points indeed to an important control which was not included in our original version. We have tested whether the establishment of SeMeCo is explained by genetic alterations of its members. We isolated the different auxotrophs from an established SeMeCo, and then we repeated the co-culture experiments as done in the original strains. The different auxotrophs isolated from SeMeCo behave like the original strains, and do not co-grow upon co-culturing. This shows that the cooperation in SeMeCo did not establish due to mutations of genomic altercations which overcome feedback inhibition, as this would be then inherited (Figure 2—figure supplement 3).

Somewhat related, is it possible to reproduce mutualistic growth from scratch, in co-cultures, using mixed populations of cells with metabotype composition similar to SeMeCos? Perhaps in conditioned SeMeCo media, since the authors suggest that "exchange interactions will readily establish once the cellular environment has acquired a critical concentration of metabolites".

This control is included Figure 2—figure supplement 4. Cooperative growth is not achieved when mixing auxotrophs at the same percentage as isolated from the self- established community. This outcome is not affected either by co-cultivating auxotrophs together in rich media prior to spotting.

To the second comment, the auxotrophs grow upon supplementation of histidine, leucine, uracil and methionine, which are the four metabolites that limit the growth of the auxotrophs, and are therefore the minimum set of intermediates that need to be exchanged. Uptake rates of these metabolites from complex media composition for both auxotrophs and prototrophs are given in Figure 1.

I do not quite understand why SeMeCos transferred to the liquid minimal medium – and at very low dilutions (Figure 3) – do not show longer lag phase as compared to their parent strain and grow normally. Do they overproduce and/or over-secrete metabolites under these conditions? Would it be expected that metabolite concentration is initially very low? Or are those initially proliferating cells prototrophs that may later start losing plasmids, re-establishing the community?

The reviewer is fully correct, the lag phase is longer in SeMeCo and the segregating strain containing the four plasmids (Figure 3). We apologize that we did not discuss the lag phase in the first version of the paper, this has been included now. When starting the segregation experiments, initially all cells are prototrophs, and hence all metabolites are initially produced by prototrophs. Once SeMeCo is established, metabolites are produced by cells depending on their auxotrophic genotype. Please see the experiments in Figure 5, that shows the uracil-supplemented SeMeCo, where prototrophs are practically depleted.

I do not get the logic behind experiments shown in Figure 1. The authors first show that many metabolites are found in extracellular space within yeast colonies. They then argue that these metabolites must be exported rather than released from lysing cells because endo- and exometabolomes of exponentially growing cells are different. I appreciate that growing cells may well export metabolites but why is this detour to exponentially growing cultures where cell lysis is negligible? How is it related to colonies where growth is restricted to a small outward layer and cell viability within a colony center is likely decreased?

We have addressed this point in detail, it’s only a control to rule out adaptive re-growth. Please see Essential Revision Point #3.*Similarly, what is the reason for including an experiment shown in 2Av? The fact that cells lose plasmids in the absence of selection is not particularly surprising.*

This is another control experiment confirming that the segregation over time follows the measured segregation rate. In fact we have been asked for this control upon presenting our results in at least three seminars, and hence would like to keep it in the paper. If this point would not have coming up repeatedly (which shows that several readers seem to expect this control), the authors fully agree with the reviewer, the additional information content of this control experiment is not very extensive.

What is the degree of cell death during community establishment? Somewhat related, what is the efficiency of SeMeCo establishment? Do all cells eventually give rise to a mutualistic community?

We have tested this in an experiment (please see Essential Revision Point 3). The number of colony forming units in an exponentially growing wild-type and SeMeCo population are the same, and correspond to the total number of cells in the SeMeCo (please see Essential Revision Point 3). This confirms a similar behaviour of SeMeCo to prototrophic yeast, that is known to have a very low number of cell death during exponential growth, in that almost every cell can give rise to a colony.

Reviewer #4:*This is an interesting paper describing how it is possible to evolve yeast communities where certain members produce specific amino acids/nucleotides and other members consume these metabolites. The authors demonstrate that this community behaves similar to wild type prototrophic strains, both in terms of specific growth rate and in response to stress. The authors present an impressive number of controls and perform a detailed analysis of the established community. So overall this is a sound paper and I do not really have any specific comments to the Results and Discussion. However, I am a little puzzled about the rationale of this work. It is not clear what we are learning in terms of new biology from the study. It is an interesting observation that both phototrophs and auxotrophs take up metabolites at the same rate, but I think this has been observed before. The authors can probably easily revise their paper to take this point into account, as I am certain that they in fact could argue for the rationale of this study.*

We thank the reviewer for the overall assessment. As detailed in the general comments and in response to Reviewer #3, we have revised the manuscript to explain better the biological implications, which are of major importance for the metabolism field, as they show that in co-growing cells, intermediate metabolism is substantially operating in a non-cell-autonomous manner. We also appreciate the comment about comparison of production and uptake rates. We have done an intensive literature research, and also to our own surprise, did not find a manuscript that would have contained data allowing us to compare the uptake rates for histidine, leucine, uracil and methionine between a prototroph and corresponding auxotroph.

References:

Momeni, B., Waite, A.J., and Shou, W. (2013b). Spatial self-organization favors heterotypic cooperation over cheating. eLife 2, e00960.

Váchová, L., Cáp, M., and Palková, Z. (2012). Yeast colonies: a model for studies of aging, environmental adaptation, and longevity. Oxid. Med. Cell. Longev. 2012, 601836.

Wintermute, E.H., and Silver, P.A. (2010). Dynamics in the mixed microbial concourse. Genes Dev. 24, 2603–2614.

[Editors' note: further revisions were requested prior to acceptance, as described below.]

Reviewer #2: *[…] That said, as I was re-reading the paper, I again started wondering why auxotrophic mutants, even when mixed at the same ratios as found in SeMeCos, fail to establish a growing population; whereas prototrophic cells that contain unstable plasmids are able to form a community with some fractions of the population showing plasmid loss that results in auxotrophy. I understand the authors' argument that feral yeasts often do not show auxotrophies because they may only rarely grow exponentially and because they may go through sexual cycles where they need to be able to survive and divide as single haploid cells. But why would mixing auxotrophs in rich medium not yield the same result as a population where the haploids are gradually formed through plasmid loss as the population grows?*

We apologize that we had not discussed the results about rich media in our manuscript: In rich media the auxotrophs certainly grow without any apparent growth defect. So as the reviewer correctly states, in rich media, the mixed auxotroph, SeMeCo and the wild type strains reveal similar growth properties. We have in depth studied the growth properties of auxotrophs in supplemented media in our previous work, (Muelleder et al., Nature Biotechnology 2012), and we have included this reference and the description. Self-establishment of the community is only required to establish growth in minimal media.

To the other point, we are fully in line with the reviewer that the limiting factor is not the ratio in which the auxotrophs are mixed to each other; the key is to allow progressive self-establishment of the community starting from the initial single cell, which is certainly the main point of the paper. We have clarified this statement in the Discussion.

Related to this is the question of what prevents such growth upon co-culturing. The answer for this comes from both Shou and Murray labs, that have demonstrated that when mutating feedback regulatory mechanisms, co-cultures can complement each others deficiency (Shou et al., 2007, Muller et al., 2014). Regulatory feedback mechanisms hence limit the co-cultures to growth. We are not questioning this fact; our results however show that the initial interpretation of this finding, that yeast would not possess sufficient import/export capacities, is not correct.

*One important potential artefact would be that after plasmid loss, the proteins encoded by the lost plasmids remain present and active in the cells for a few generations. It is well known that protein carry-over can persist for more than 5 doublings, so in principle, it could be possible that the population contains "zombies" – cells that are able to survive and even divide for a few rounds, but whose lineage is eventually bound to disappear when the limiting proteins become too diluted because of divisions and protein turnover…. This could also explain why at least a fraction of the population remains prototrophic.*

We can fully exclude this being a problem for the interpretation of our experiments for the following reason: We are not monitoring the plasmids themselves, but the auxotrophic phenotype of the cells. For this we replicate a minimum number of 200 colonies on six media types (~1200 spots per replicate); and if a cell can grow, it counts as a prototroph, if not, as an auxotroph. Even if it would have lost the plasmid a cell division before that (but is still biochemically competent by possessing the biosynthetic enzyme) it is still recognised as a prototroph as long as it physiologically is one (and we fully agree with the reviewer, this is what matters). This is in fact the huge advantage of using replica plating over microscopy or PCR based techniques that would only give an indirect measure of auxotrophy over the presence of absence of the gene.

Second, there is a control in respect to this in the manuscript. We have compared the segregation rate prediction with the actual appearance of auxotrophs on YPD media (Figure 2). The results are overall nicely in agreement, implying that if pseudo- prototrophic cells with protein carry-over do exist, they are not significantly influencing overall population composition.

Finally, the maintenance of the prototrophs is explained by the uracil requirement; once this metabolite is supplemented, the remaining content of prototrophs is dramatically reduced (Figure 5).Reviewer #3 [abridged]: *I am fine with the revision – it is a considerably improved and focused version of the manuscript. There are several instances where the text will benefit from further edits. I will point out a few: The authors mention that they "re-isolated individual prototrophs from the established colonies…". I am somewhat confused – is this a typo and it should be "individual auxotrophs", since the authors attempt and fail to co-culture them later? Or do the authors mean that culturing prototrophs for 48 hour in supplemented medium (liquid? solid?) may induces plasmid loss? Either way, it could be worth clarifying this point.*

We apologize that we have misleadingly written this paragraph. What we meant with “individual prototrophs” is a strain prototrophic for methionine, uracil, leucine and/or histidine, but that it is auxotroph for the other three markers. The reviewer is correct that the appropriate would have been “individual auxotrophs”. We now have simplified the paragraph.

*In the subheading “SeMeCos reveal composition dynamics in response to nutritional changes”, the authors state that: "These findings are hence consistent with the notion that metabolic cooperation prevents population intermixing, so that cooperation partners within the colony retain close proximity to prevent diffusion of the exchanged metabolite". Yet, the authors also show that metabolite exchange is not constrained by colonial growth and does continue in liquid cultures. I wonder if the authors could add a sentence or two to discuss this point.*

We have expanded this section: There is literature saying that in a metabolically cooperating community, metabolite exchanging cells keep physical proximity to each other, and that this would be a condition for metabolic cooperation. The reviewer is right, we see proximity reflected within the colony, not however in liquid culture. The liquid culture experiments show spatial structure is not a basic condition for metabolic cooperativity.